# Lost in the Maze: Overcoming Context Limitations in Long-Horizon Agentic Search

## Abstract

Long-horizon agentic search requires iteratively exploring the web over long trajectories and synthesizing information across many sources, and is the foundation for enabling powerful applications like deep research systems. In this work, we show that popular agentic search frameworks struggle to scale to long trajectories primarily due to context limitations—they accumulate long, noisy content, hit context window and tool budgets, or stop early. Then, we introduce SLIM (**S**imple **L**ightweight **I**nformation **M**anagement), a simple framework that separates retrieval into distinct search and browse tools, and periodically summarizes the trajectory, keeping context concise while enabling longer, more focused searches. On long-horizon tasks, SLIM achieves comparable performance at substantially lower cost and with far fewer tool calls than strong open-source baselines across multiple base models. Specifically, with o3 as the base model, SLIM achieves 56% on BrowseComp and 31% on HLE, outperforming all open-source frameworks by 8 and 4 absolute points, respectively, while incurring 4–6x fewer tool calls. Finally, we release an automated fine-grained trajectory analysis pipeline and error taxonomy for characterizing long-horizon agentic search frameworks; SLIM exhibits fewer hallucinations than prior systems. We hope our analysis framework and simple tool design inform future long-horizon agents[1].

## 1 Introduction

Long-horizon agentic search involves performing searches over long trajectories and reasoning over many sources, and requires powerful systems that can explore diverse sources and leverage tools effectively. The ability to reason over long trajectories serves as the foundation for exciting applications such as deep research (OpenAI, 2025; Google, 2025; xAI, 2025). Due to its immense potential in solving complex tasks, long-horizon systems have been a key focus in the community, eliciting the development of many proprietary and open-source frameworks. Among open-source systems, HuggingFace Open Deep Research (Roucher et al., 2025) and GPT Researcher (Elovic, 2023) opt for complex multi-agent orchestration while SEARCH-O1 (Li et al., 2025b) uses a single agent. However, despite the numerous approaches, they still fail in complex long-trajectory settings, and there are no systematic approaches to analyze their trajectories and identify the failure modes.

In this work, we first analyze existing frameworks by examining their trajectory outcomes on BrowseComp (Wei et al., 2025), a challenging long-horizon agentic search benchmark. Our analysis shows that these frameworks still struggle with long-trajectory tasks, failing on more than 50% of the samples—most of the failures are due to hitting the context window limit, running out of tool budget, or stopping too early.

We attribute these failure modes to poor context management that can fill the context window with noisy information that derails long search trajectories. The limited context restricts the number of turns in each trajectory, resulting in incomplete information gathering. To overcome these limitations, we design SLIM (**S**imple **L**ightweight **I**nformation **M**anagement), a framework with three simple yet powerful components—search, browse, and summarization—that effectively manage the context size of long-horizon systems. The simple tool design allows LLMs to interleave searching for diverse information and browsing promising pages without spending unnecessary tool calls on noisy search

---

[1]Our code will be made publicly available.

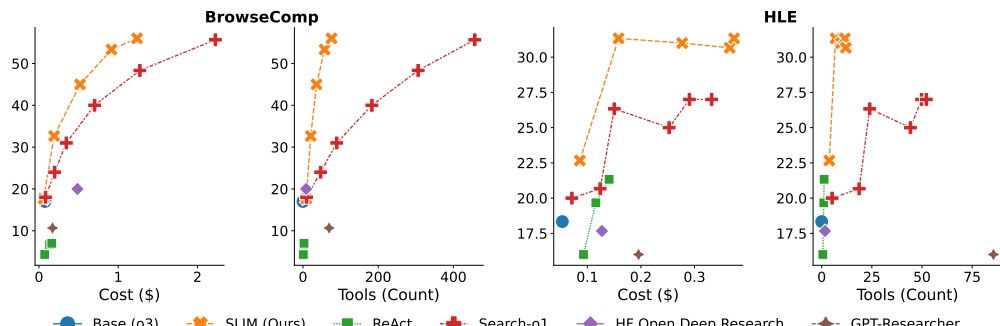

Figure 1: With o3 as the base model, SLIM achieves better performance than existing frameworks on both BrowseComp and HLE while using more than 4-6x fewer tool calls and lower overall costs, which account for LLM token usage and tool costs.

results. Furthermore, the summarization module acts as a general-purpose context manager that can reduce long trajectories into more condensed summaries. These design choices combine to allow the system to scale to longer trajectories while maintaining a concise context and reduced tool costs. Under a comparable cost budget, with o3 as the base model, SLIM significantly outperforms the previous best open-source frameworks by 8 and 4 absolute points on BrowseComp and HLE, respectively, while requiring only 15-25% of the tool calls (Figure 1).

Finally, we introduce an automated trajectory-level analysis pipeline that provides fine-grained insights into long-horizon frameworks. To characterize mistakes made by these systems, we develop an error taxonomy identifying common failure modes. Our analysis reveals that SLIM's advantage stems from its robustness to failure modes such as hallucinations and unfocused and generic searches. We hope our analysis pipeline, error taxonomy, and careful design choices in SLIM can serve as a foundation for understanding and improving long-horizon agentic search systems.

## 2 PRELIMINARIES: LONG-HORIZON AGENTIC SEARCH

Previous information-seeking tasks, such as open-domain question answering, are simpler, as they typically involve factoid questions that are easy to answer with a single source (Joshi et al., 2017; Kwiatkowski et al., 2019; Petroni et al., 2021). As a result, these tasks can be mostly solved with static retrieval-augmented generation (RAG) systems that leverage at most a few retrieval steps (Lewis et al., 2020; Izacard et al., 2023; Shi et al., 2024), and do not showcase the challenges of realistic, long-horizon agentic search settings. In contrast, we study long-horizon tasks with complex queries that require extensive searches to gather the necessary information and reasoning over different sources to synthesize the answer. In this section, we formalize the task, describe the datasets for studying long-horizon agentic search, and review some previous long-horizon systems.

### 2.1 TASK FORMULATION

We formalize long-horizon agentic search tasks as follows: given a query $q$, a corpus of documents $\mathcal{D}$, the system needs to perform a sequence of tool calls to find relevant information from $\mathcal{D}$ and output a final answer $o$, which is checked against the annotated groundtruth answer $a$. A critical component of the system is the design of its tools and how it interacts with the corpus; each tool is a function $\mathcal{T}_i(x) \to y$ that maps arbitrary system-generated inputs $x$ to arbitrary outputs $y$.

Furthermore, agentic systems are often controlled by a tool budget $T$, the maximum number of tool calls they are allowed to use in any trajectory. The tool budget $T$ also corresponds to the maximum number of turns in a trajectory, as each turn corresponds to one tool call[2]. Thus, how to manage the input context to the underlying LLM across many tool uses and turns is another critical design choice in long-horizon systems. Finally, the final step where the system outputs its final answer does not count towards the tool budget.

---

[2]Some architectures, such as the CodeAgent (Wang et al., 2024) used in HF-ODR, allow for parallel tool calls in one step (e.g., using for loops), but we found that the models we tested do not use this capability.

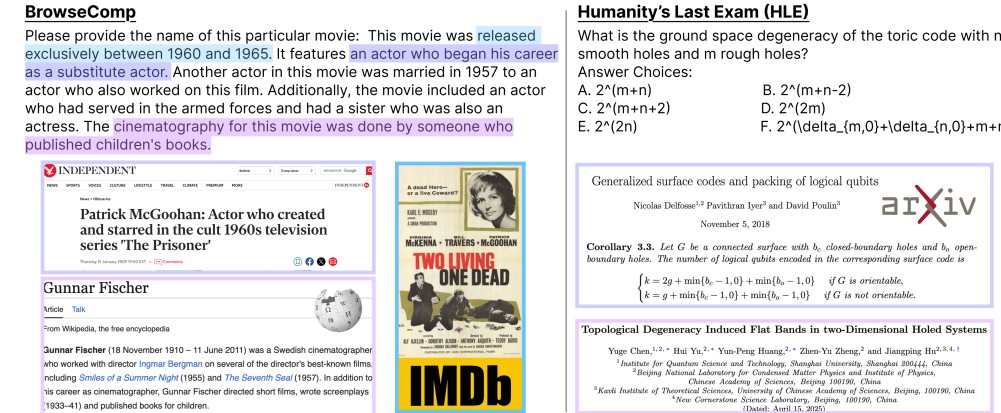

Figure 2: Example queries and their relevant documents for BrowseComp (Wei et al., 2025) and HLE (Phan et al., 2025).

In long-horizon agentic search settings, the web is most often used as the corpus $\mathcal{D}$ due to the diversity and complexity of the queries, and each document $d_i = (u_i, t_i, c_i)$ comprises a URL, title, and content. In practice, long-horizon systems typically use search engines $\mathcal{R}(q) \rightarrow \{(u_i, t_i)\}_1^n$ to obtain a list of $n$ web pages with their titles and URLs most relevant to the search query $q$. Furthermore, a scraping operation $\mathcal{C}(u_i) \rightarrow c_i$ is necessary to obtain the full content of any URL as search engines only provide a list of URLs, but scraping is slow and noisy in practice.

In traditional QA settings, since the retrieval tool only needs to be called once due to the simplicity of the queries and the small size of the corpus (i.e., Wikipedia), retrieval returns the full list of documents and their contents $\mathcal{R}_{\text{wiki}}(q) \rightarrow \{(t_i, c_i)\}_1^n$. As a result, many long-horizon systems follow a similar design, where the retrieval tool is a single search engine call followed by scraping all returned URLs. However, the complexity of long-horizon agentic search requires many tool calls to gather the necessary information (Li et al., 2025b; Jin et al., 2025b). As we demonstrate empirically later, this naive tool design leads to severe context limitations, where the system is overwhelmed by long, noisy content, motivating the design of more efficient tool interfaces for long-horizon systems.

## 2.2 DATASETS

We select two datasets with naturally difficult queries that require long-trajectory searches and verifiable answers, which ensures the reliability of subsequent analyses. For evaluation, we sample a random subset of 300 instances from each dataset due to the high costs of running long-horizon systems. An example query from each dataset is shown in Figure 2.

**BrowseComp** (Wei et al., 2025) consists of challenging queries targeting hard-to-find information. BrowseComp tests one of the core capabilities of long-horizon systems—the ability to exhaustively search the web over long trajectories and collect the necessary information. These queries were rigorously validated by humans to require $> 10$ minutes of searching on the open web. As a result, BrowseComp is extremely challenging for long-horizon systems.

**Humanity's Last Exam** (HLE; Phan et al., 2025) tests across multiple domains and often requires domain-specific knowledge and reasoning skills. These expert domains span across a wide range of topics, such as biology, mathematics, and physics. HLE tests the ability of long-horizon systems to leverage the web to find helpful information that can aid reasoning-heavy problems. These questions are rigorously vetted by domain experts, and most existing systems fail to achieve high accuracy. We use the text-only subset to allow for evaluation of text-only systems.

Table 1: Comparison of SLIM with existing frameworks. In contrast to single-agent works that bundle search and browsing search results into *one* retrieval tool, we separate it into two distinct tools.

| Framework | Architecture | # Tools | Tools | Input to LLM Context | Summarization |
|---|---|---|---|---|---|
| REACT | Single-agent | 1 | Retrieval | All search results | - |
| SEARCH-O1 | Single-agent | 1 | Retrieval | All search results | Retrieved content |
| HF-ODR | Multi-agent | 11 | Search, Browse, Python, ... | Selected search results | Search agent result |
| GPT-R | Multi-agent | 1 | Retrieval | All search results | Retrieved content |
| SLIM (ours) | Single-agent | 2 | Search, Browse | Selected search results | Task trajectory |

## 2.3 Existing Approaches

We briefly describe some popular approaches to agentic search, ranging from simple single-LLM frameworks to complex multi-agent systems. We summarize the differences between these frameworks in Table 1; more details are in §A.1.

**REACT** (Yao et al., 2023) is a simple framework that allows an LLM agent to alternate between thinking and acting, allowing tool calling across many turns. Following the original work, our implementation gives the LLM access to a single retrieval tool—given a query, the tool returns a list of top 10 results along with their web contents. All search results are then concatenated to the agent's context for subsequent steps. When the LLM chooses not to use the search tool, the final output is used for evaluation. Our experiments vary the maximum number of turns in each trajectory.

**SEARCH-O1** (Li et al., 2025b) builds upon REACT with an additional "reason-in-document" step, where an LLM summarizes the search results and their contents before appending the results to the agent's input context. Although the summary step reduces context length for the main LLM compared to REACT, this approach still uses many scraping operations in each search step (one for each search result), and summarization incurs a large amount of LLM token usage.

**HuggingFace OpenDeepResearch** (HF-ODR; Roucher et al., 2025) leverages a hierarchical structure consisting of a manager agent and a search agent. The manager agent calls the search agent to perform detailed searches. The search agent iteratively interacts with a search engine, a browser, and other tools (detailed in §A.2), and returns a summary of its searches. The manager agent may use the summary to issue more queries or output a final answer. We use the default settings, which fixes the maximum number of turns $T = 20$ for the manager and search agent.

**GPT-Researcher** (GPT-R; Elovic, 2023) is a complex multi-agent system where each agent has distinct roles: a research conductor that orchestrates the search process, a report generator that creates the report, a context manager that summarizes search results, and a source curator that selects relevant sources from scraped pages. The system uses a deep researcher agent that acts as a search tree node, spawning multiple children nodes with these same components. We use the default setting, which fixes the depth of the search tree $= 2$ and the breadth of search at each depth $= 4$.

## 3 Failure Modes of Existing Approaches

Despite recent progress, we still know little about how individual components in these systems perform, or fail. To study behavior on long-horizon tasks, we focus on BrowseComp, which naturally induces extended, multi-step search trajectories. For this task, the final outcome can reveal the overall performance of each framework as well as its relationship with the context window limitation and tool budget constraints. For this analysis, we let the framework run up to a fixed number of turns and output an answer. We categorize the final outcome in Table 2.

For this analysis, we consider different tool budgets for REACT and SEARCH-O1, and use the default 20 turns for HF-ODR. We observe that context window limitations and tool budgets are the main bottlenecks for existing approaches in Figure 3, and each framework exhibits distinct patterns.

Specifically, REACT often hits the context window limit over a long trajectory due to the large amount of text returned by each search call. As a result, it cannot effectively scale to long trajectories and

Table 2: Categorization of different search outcomes and their descriptions.

| Outcome | Description |
|---|---|
| Correct | The system outputs the correct answer |
| Exceed context | The system exceeds LLM's context window, falling back to not using any tools |
| Exceed budget | The system exceeds the tool calling or iteration budget |
| Early stopping | The system outputs an incorrect answer before reaching the iteration budget |
| No tool used | A special case of early stopping where the system does not use any tools |
| Misc. error | Due to uncontrollable factors (e.g., API content filters) the system outputs an error message |

Figure 3: Each framework exhibits distinct outcome trends—REACT predominantly runs out of context window, while SEARCH-O1 is often limited by the tool budget (T). We exclude GPT-R due to its predefined workflow—the outcome can only be either correct or incorrect.

make full use of its tool budgets. SEARCH-O1 failure cases are almost entirely due to exceeding the tool budget, which suggests increasing the tool budget may potentially lead to better performance. However, such an increase is non-trivial without incurring a significant amount of cost—each retrieval step in SEARCH-O1 involves scraping all search results, even though only a fraction of these results are relevant, leading to a large amount of LLM token consumption during the summarization step.

Finally, we observe that HF-ODR often prematurely terminates due to the manager agent's inability to leverage its search agent across multiple steps. Furthermore, HF-ODR is the only framework that do not use any tools in a significant percentage of the trajectories (10%), suggesting that complex prompt-engineered workflows may be prone to reducing the tool calling capabilities of the base model. The root cause of these failure modes is poor context management—exceeding context and tool budgets, or stopping too early. In the next section, we explore how to substantially improve agentic search frameworks through better context management.

## 4 OUR FRAMEWORK: SLIM

A key takeaway from our analysis is that long-trajectory tasks require **scaling up the number of turns and tool calls while keeping the context concise to avoid hitting the context window limit**. Specifically, search results are often noisy and irrelevant to the answer, so filling up the context with content from all search results can lead to noisy context and unnecessary tool costs. Motivated by these observations, we introduce SLIM (**S**imple **L**ightweight **I**nformation **M**anagement) with two key principles: (1) using simple and flexible tools for LLMs to interact with, and (2) minimizing the amount of noisy information presented to the model and keeping the context concise during exploration. An overview of SLIM in comparison to existing frameworks is shown in Figure 4.

Concretely, SLIM adopts three simple yet powerful components—search, browse, and summarization—to effectively manage the context and scale the number of turns.

**Search tool $\mathcal{R}$.** As the main vehicle for exploring the web, SLIM uses a simple and fast interface for the search tool. Specifically, the search tool only returns the top $k$ search results from a search engine, where each search result consists of a title, a URL, and a short snippet of its content. A crucial difference from previous frameworks is that previous work often bundles the search and browse functionality and returns the full content for all search results, and relies on the main LLM to discern relevant context. In comparison, our search tool only returns a short snippet of each result, keeping the output concise and avoiding wasting context and tool calls on irrelevant content.

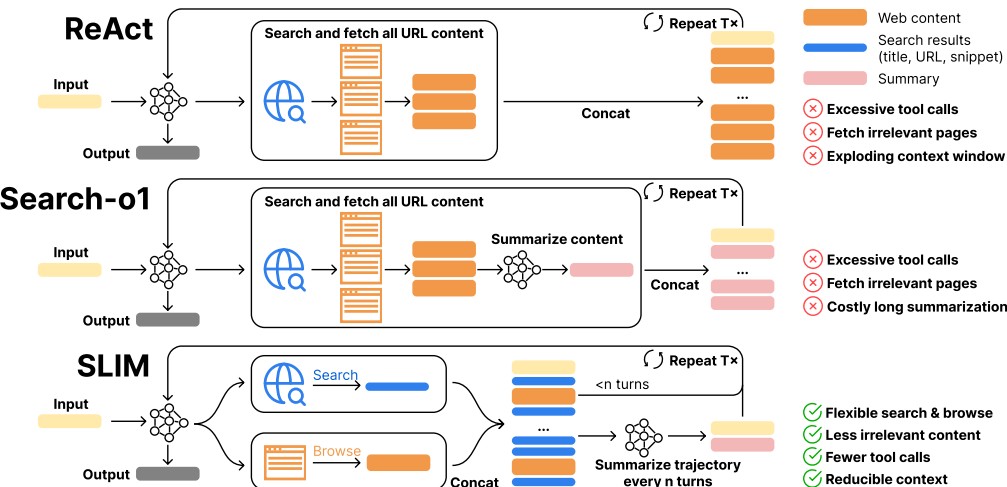

Figure 4: Compared to REACT and SEARCH-O1, the cooperation between search, browse, and summarization modules allows SLIM to accumulate shorter contexts and less noisy information after exploring the same amount of searches.

**Browse tool $\mathcal{B}$.** Our browse tool is designed to complement the search tool by allowing the LLM to dig deeper into promising search results. Specifically, the browse tool $\mathcal{B}(u, q) \to \max_{c_i \in c} \mathrm{sim}(c_i, q)$ returns the most relevant section of the content $c$ from the URL $u$ to the query $q$. Notably, this design enables the LLM to select the most relevant search result and choose a subset of the content that best matches the specific information it is looking for. As a result, our browse tool is significantly more efficient and cheaper than previous frameworks that exhaustively browse all search results in terms of both the scraping operations and the amount of new tokens introduced to the context.

**Summarization module $\mathcal{S}$.** Despite the brevity of each tool response, agent context inevitably grows as it explores over a long horizon of searches. To maintain a concise context while retaining the effective exploration history, we introduce a summarization module that periodically compresses the LLM context. We find a simple heuristic sufficient: we summarize the entire conversation history after every $n$ turns of tool calls and replace the trajectory with the summary. This crucially differs from previous works where summarization is solely applied to search results at each turn.

Finally, we combine these components into a single framework by allowing the underlying LLM to call either the search or the browse tools at every turn. Then, the summarization module compresses the entire conversation every $n$ turns to reduce the amount of noise. Our implementation uses Google[3] as the search tool, crawl4ai[4] as the browse tool, and the same LLM as the agent model for summarization. More details, an example trajectory, and ablations on the search tool, browse tool, and summarization module are shown in §A.4.

## 5 RESULTS

We use o3, o4-mini, and Claude-4-Sonnet as our base models. For each instance, we evaluate the system's performance as well as the number of tool calls and tokens used. The number of tool calls is the sum of the search API and browse/scraping operations. For the number of tokens, we take a weighted sum of the LLM input and output tokens across all turns. We exclude cached input tokens from the total tokens count since practical systems are typically implemented with caching mechanisms in long-trajectory tasks with shared context. For each dataset we report results averaged over all instances. More details on the experimental setup can be found in §A.5.

We present the main results with o3 as the base model in Table 12. Under the same cost, SLIM achieves significant improvements over SEARCH-O1, the best performing open-source framework,

---

[3] https://serper.dev/
[4] https://github.com/unclecode/crawl4ai

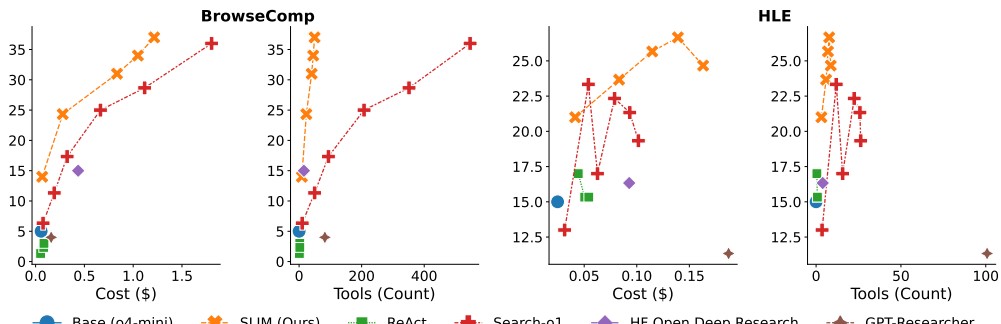

Figure 5: With o4-mini as the base model, SLIM consistently outperforms other baselines on BrowseComp while using fewer tool calls and lower overall costs. On HLE, SLIM can achieve overall higher performance and use fewer tool calls.

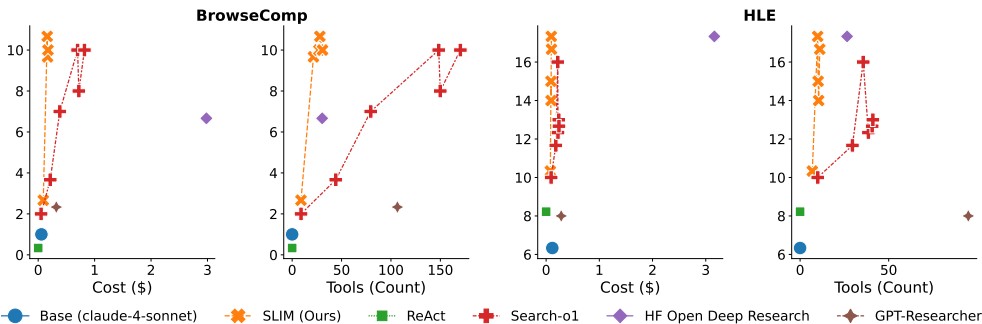

Figure 6: With Claude-4-Sonnet as the base model, SLIM consistently outperforms other baselines on BrowseComp while using fewer tool calls and lower overall costs. On HLE, SLIM can achieve overall higher performance and use fewer tool calls.

by 8 and 4 points on BrowseComp and HLE, respectively. The difference is more pronounced when controlling for cost: on BrowseComp, SLIM can scale to 150 turns while using less total cost and reaching higher performance than the corresponding SEARCH-O1 setting (50 turns). Furthermore, SLIM uses significantly fewer tool calls—less than 25% of the tool calls used by SEARCH-O1— suggesting that SLIM can leverage tools much more efficiently. The performance-cost comparisons of these systems are shown in Figure 1, and the detailed numbers and comparisons are shown in Table 12. We also conduct statistical tests to compare the performance of SLIM with the baselines, as shown in Table 13.

We also show results with different base models—o4-mini in Figure 5 and Claude-4-Sonnet in Figure 6. SLIM consistently achieves the highest performance across these models and all datasets compared to other frameworks, suggesting that our simple design generalizes well to models of different sizes and training strategies. Furthermore, our effective context management also results in fewer tool calls and often lower overall costs compared to the baselines. SLIM also shows consistent trends across all three base models whereas certain frameworks only work well under certain models; for instance, HF-ODR only achieves competitive performance with Claude, where the engineered prompts are more effective. Overall, this is strong evidence that SLIM serves as an effective framework for long-horizon tasks. We show tables with full results and ablations on the baselines in §A.6.

## 6 FINE-GRAINED TRAJECTORY-LEVEL ANALYSIS

### 6.1 TRAJECTORY-LEVEL ERROR TAXONOMY

To understand how SLIM improves over other systems at a deeper level, we extend the analysis beyond the task outcome, and focus on characterizing the mistakes that a system makes over the course of its

**Question:** Provide the birth name of a certain individual:
1. hired for a coordinator position in 2012 and later promoted
2. has a child that was born in the United States.
3. released their debut single between 2010 and 2015.

**Groundtruth Answer:** Nicholas Munene Mutum (a Kenyan actor) ✓

**Search queries:**
1: debut single 2012 filipina actress model business administration
2: "child born in the United States" singer actress
3: "promoted to manager" "debut single" 2014
...
19: Filipina actress gave birth in the United States 2015
20: Filipina actress debut single 2013
...
48: "marketing coordinator" 2012 Philippines
49: "children born in the United States" actress "Philippines"

**(1) Unfocused searches**: overly generic queries that do not narrow down search space

**(2) Confirmation bias**: ≥50% search queries focus on a wrong candidates due to early noisy signals.

**Search results:**
1: wikipedia/Maja_Salvador, imdb/Filipina Beauty
2: timenote/Virginia_Weidler, wikipedia/Sharon_Pierre-Louis
...
20: timenote/Virginia_Weidler, wikipedia/Maja_Salvador
21: wikipedia/Nick_Mutuma ✓

**(3) Inefficient search**: search repetitive information/URLs

**(4) Answer ignored**: correct answer found in trajectory but not used

**Example Output 1:**
Explanation: I was unable to reliably identify...
Exact Answer: Unable to determine from the information available

**Example Output 2:**
Explanation: Angeline Quinto satisfies every clue:
1. Angeline is a Filipino singer with a child born in the US.
2. Angeline's debut single was released in 2012.
3. Angeline was promoted from coordinator to manager at 1FMs

**(5) Abstention**: do not attempt to answer a question.

**(6) Hallucination: generated statements are not supported by contents from the trajectory.**
Cross check with search results →
2/4 unsupported statements

Figure 7: Examples of each trajectory-level failure mode on a BrowseComp sample.

long search *trajectories*. To this end, we first develop a shared taxonomy of common failure modes by manually examining individual trajectories from the compared systems on BrowseComp. We present examples of each failure mode in the taxonomy in Figure 7, and detailed definitions in §A.3. Our taxonomy covers possible failure modes for long-horizon search agents in the information gathering process (e.g., unfocused searches, confirmation bias, and inefficient search) as well as the answer synthesis stage (e.g., ignoring the answer, abstention, and hallucination).

Based on the taxonomy, we develop an automated error analysis pipeline that annotates each trajectory with the failure modes using a mix of rule-based heuristics and LLM-as-a-judge approaches. Our pipeline carefully examines all parts of each trajectory—the search queries and results, the browsed contents, and the final answer—to identify the failure modes. We describe the pipeline more in §A.3.

## 6.2 ANALYSIS OF TRAJECTORY-LEVEL FAILURE MODES

For fair comparison, we analyze all frameworks under a similar cost budget[5]. For each framework we choose the setting with the closest cost to SLIM with tool budget $T = 150$, according to Table 12. The distribution of trajectory-level errors are shown in Table 3, where we show the percentage of correct answer and each failure mode across all samples. We first observe that SLIM's advantage in performance could be attributed to the notably reduced hallucination rate compared to other frameworks. This is likely due to the fact that SLIM can choose what URLs to browse based on the search results, allowing it to reduce the amount of noise in the context. In contrast, the other frameworks observe significantly higher hallucination rates compared to SLIM, suggesting that they often resort to their parametric knowledge to answer the question when they cannot find the correct answer through tool calls.

Moreover, SEARCH-O1 and SLIM observe higher percentages of answer ignored than other frameworks. One explanation is that these frameworks tend to encounter more search results across their longer trajectories, which leads to a higher chance of finding the answer, but also a higher chance of ignoring it. In contrast, REACT and HF-ODR do not scale well to longer trajectories, which means they are unlikely to encounter the correct answer. Our analysis reveals that a promising direction for

---

[5]We exclude GPT-R because their implementation do not return the contents of the search results.

Table 3: The percentage of trajectory over all samples that observe each failure mode. For hallucination only, we report the percentage of hallucinations for samples that ends with an incorrect answer and do not abstain.

| Framework | Turn Budget | Correct | Confirm Bias | Unfocused Search | Inefficient Search | Abstention | Answer Ignored | Hallucinate |
|---|---|---|---|---|---|---|---|---|
| REACT | 10 | 7.0 | 9.3 | 44.0 | 3.9 | 1.0 | 0.7 | 56.7 |
| SEARCH-O1 | 50 | 48.3 | 9.3 | 33.7 | 7.2 | 4.3 | 26.0 | 46.8 |
| HF-ODR | 20 | 20.0 | 6.7 | 58.7 | 43.9 | 32.3 | 1.7 | 96.2 |
| SLIM | 150 | 56.0 | 9.7 | 34.0 | 7.6 | 27.7 | 30.7 | 19.0 |

improving long-horizon agentic search frameworks is to enable language models to better identify the correct answer from long trajectories.

Notably, despite the improvements on hallucination, SLIM still suffers from high abstention rates, and is more prone to ignoring the groundtruth answers. We leave these improvements to future work, and hope that our trajectory-level analysis can be a useful tool for improving long-horizon systems in more interpretable and concrete ways.

## 7 RELATED WORK

**Deep research.** Recently, the community has taken great interests in deep research systems due to their potential to solve complex tasks—there have been efforts across both industry (OpenAI, 2025; Google, 2025; xAI, 2025; Nguyen et al., 2025) and open-source communities (Wu et al., 2025a; Du et al., 2025; Sun et al., 2025, *inter alia*). They are often evaluated through long-horizon search trajectories tasks that also require complex reasoning (Wei et al., 2025; Phan et al., 2025). Other benchmarks evaluate the long-form generation capabilities of systems (Du et al., 2025).

Furthermore, between the opaque proprietary systems and increasingly complex open-source systems, there is little understanding on the underlying behavior of long-horizon systems and how they fail in practice. In this work, we aim to fill this gap by introducing an error taxonomy for long-horizon systems and an automatic error analysis pipeline. We design our automated analysis pipeline to conduct fine-grained analysis across a search trajectory, while previous works study more general multi-agent interaction (Pan et al., 2025; Deshpande et al., 2025). The two approaches, general and specific, are complementary to each other in gaining a better understanding of agentic systems. Finally, in contrast to existing open-source approaches that are growing increasingly more complex, we show that a simple approach with carefully designed tools can achieve better performance with fewer tool calls.

**Reinforcement learning for long-horizon systems.** There have been considerable efforts in improving search agents through reinforcement learning (Li et al., 2025c; Zheng et al., 2025; Chen et al., 2025; Li et al., 2025a; Wu et al., 2025b, *inter alia*). A popular approach is to synthetically generate question-answer pairs that require long-horizon search trajectories (Xia et al., 2025; Tao et al., 2025). Other works focus on comparing different training objectives (Jin et al., 2025b;a). However, critical analysis of the error modes and comparison of different frameworks are still lacking.

## 8 CONCLUSION

In this work, we propose SLIM, a simple yet effective long-horizon agentic search framework that addresses context limitations prevalent in existing systems. We show that SLIM consistently achieves the highest performance across different base models and datasets compared to other frameworks while using fewer tool calls and lower overall costs, suggesting that our simple design enables better long-horizon agentic search.

We then develop an automated error analysis pipeline to characterize the failure modes of long-horizon systems. Our analysis shows that SLIM is more resistant to failure modes such as hallucination. We hope our framework and analysis pipeline can serve as a useful tool for the community to understand and improve long-horizon agentic search systems.

ETHICS STATEMENT

This work studies the behavior of long-horizon agentic search systems, and how to improve them through better design choices. Although there are no direct ethical concerns, we acknowledge that the web and LLMs are complex systems that can be used for harmful purposes.

REPRODUCIBILITY STATEMENT

All of our experiments were conducted between August 2025 and October 2025, and we release the output files for all of our experiments. Although we release the code and results publicly, the stochastic nature of LLMs and search engines makes it difficult to exactly reproduce the results shown. While we try to control for this by running all experiments around the same time, there may still be slight differences in the results (e.g., same search query may yield different search results due to search engine updates and indexing).

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

# A    APPENDIX

## A.1    EXISTING FRAMEWORKS

**REACT** (Yao et al., 2023) is a simple framework that allows an LLM agent to alternate between thinking and acting. This framework allows the agent to use tool calls across many turns. Following the original work's knowledge-intensive task settings, our implementation gives the LLM access to a single search tool—given a query, the tool returns a list of top 10 search results, from a search engine, along with their web contents. The search results are then concatenated and appended to the agent context for subsequent steps. When the LLM chooses not to use the search tool, the final output is used for evaluation.

In our implementation, we vary the maximum number of turns in each trajectory from 1 to 10. Consistent with SLIM, we use Google as the search engine, accessed through the Serper API, which returns a list of top 10 search results. Each search result contains a title, a URL, and a short snippet of the content. After obtaining the top 10 search results, we emulate previous RAG approaches by scraping all search result URLs and concatenate their content. Similar to SLIM, we use crawl4ai to scrape web pages. We truncate each scraped document to at most 10,000 characters, which corresponds to roughly 1,000 tokens.

We notice that REACT often hits the context window limit as the retrieval results are often too long. When the LLM API call fails due to the context window limit, we fallback to not using any tools and just ask the base LLM to answer the question. As a result, we only experiment with up to 10 turns, where the framework already falls back to not using any tools for most queries. A sketch of the framework is shown in Alg. 1.

**SEARCH-O1** (Li et al., 2025b) builds upon REACT with an additional "reason-in-document" step, where an LLM summarizes the list of top 10 search results and their contents before appending the results to the agent's input context. Although the summary added to the agent context is relatively short compared to the full search result, this approach still uses a large amount of browse calls in each search step, and the summarization steps incur a large amount of LLM token usage. In our setting, we vary the maximum number of turns in each trajectory from 1 to up to 100 turns.

Similar to REACT, the retrieval tool at each step consists of a single Serper API call, followed by multiple scraping operations. We adopt the code from the original implementation[6], which uses BeautifulSoup[7] to scrape the search result URLs. In this implementation, the scraping operation will extract part of the web content that best matches the short snippet returned by the search engine. The matching is done by simply computing the F1 scores between the snippet and sentences in the web page. Subsequently, the context is filled up with at most 2,500 characters from the web page. Then, all context from the search results are concatenated and appended to the agent context for the summarization step.

It is important to note that the scraping operation is relatively expensive due to the network latency, resulting in long running time for the framework. A sketch of the framework is shown in Alg. 2.

**HuggingFace OpenDeepResearch** (HF-ODR; Roucher et al., 2025) leverages a hierarchical structure consisting of a manager agent and a search agent. The manager agent calls the search agent to perform detailed searches, and the search agent iteratively interacts with the search engine and a simulated browser to gather information. When the search agent concludes its searches, it generates a summary of its searches and returns it to the manager agent. The manager agent may use the summary to issue additional queries or output the final answer. Furthermore, another key feature of HF-ODR is its access to additional tools, such as a Python interpreter. We use the default settings[8], which fixes the maximum number of turns for the manager and search agent to be 20. A sketch of the framework is shown in Alg. 3. Specific descriptions of each tool can be found in Section A.2.

---

[6]https://github.com/RUC-NLPIR/Search-o1
[7]https://beautiful-soup-4.readthedocs.io/en/latest/
[8]https://github.com/huggingface/smolagents/tree/main/examples/open_deep_research

**GPT-Researcher** (GPT-R; Elovic, 2023) is a complex multi-agent system where each agent has distinct roles. Specifically, the system consists of a researcher conductor that orchestrates the search process, a report generator that generates the final report at the end of the search process, a context manager that summarizes search results, and a source curator that selects relevant sources from scraped web pages. Finally, GPT-R uses a deep researcher agent that acts as the node of a search tree, where each node is able to spawn multiple child nodes, each of which is a system with the previously described components. We use the default settings of the framework[9], which fixes the depth of the search tree to be 2 and the breadth of search at each depth to be 4. A sketch of the framework is shown in Alg. 4.

**Other frameworks.** There are many recent works on agentic search systems and memory-management frameworks (Gangi Reddy et al., 2025; Xu & Peng, 2025; Belcak & Molchanov, 2025). We chose the most popular open-source agentic search and deep research systems for comparison. These systems also span both simple single-agent and complex multi-agent systems, which we believe serve as a representative and fair group of baselines for the paper. Due to the high cost and long runtime of agentic systems, we only evaluate the representative baselines. Although there are explicit memory-management frameworks, we find that existing summarization models already do something similar to memory-selective mechanisms through qualitative analysis. In the example trajectory we show in Figure 8, the model summarizes the trajectory into several bullet points, such as "Investigation and findings so far", "Current hypothesis", and "Needed next". The resulting summary is similar to many memory-selective mechanisms that only retain relevant facts to the current query. Thus, we find that allowing the model to compress the full trajectory naturally filters out irrelevant information while achieving simplicity and avoiding over-prompt-engineering.

---

**Algorithm 1:** ReAct

---

**Data:** Task input $x$, LLM $\theta$, maximum number of turns $T$
**Function** *search(q)*:
     **return** *(title_i, url_i, snippet_i)$_{i=1}^{k}$*;
**Function** *browse(u, q)*:
     $D \leftarrow \text{scrape}(u)$;
     **return** $D[: 10000]$;
**Result:** Task output $y$
Turn $t \leftarrow 1$;
Context $C \leftarrow \{x\}$;
$\mathcal{T} \leftarrow \{\text{search}\}$;
**while** $t < T$ **do**
     $o_t \leftarrow \theta(C; \mathcal{T})$ ;               /* LLM may only call the search tool */
     **switch** $o_t$ **do**
         **case** *search* **do**
             $R \leftarrow \text{search}(o_t)$ ;              /* Perform search */
             $C \leftarrow C \cup \{o_t\}$;   /* Browse every search result and append */
             **for** $(t_i, u_i, s_i) \in R$ **do**
                 $C \leftarrow C \cup \text{browse}(u_i, s_i)$
         **case** *Final Answer* **do**
             **return** $o_t$;
     $t \leftarrow t + 1$;
**return** $\theta(C; \text{final answer})$;

---

## A.2 HUGGINGFACE OPEN DEEP RESEARCH TOOLS

HF-ODR is a hierarchical framework that consists of a manager agent and a search agent. The manager agent has access to the following tools:

---

[9]https://github.com/assafelovic/gpt-researcher

---

**Algorithm 2:** Search-o1

---

**Data:** Task input $x$, LLM $\theta$, maximum number of turns $T$, summary interval $n$

**Function** *search(q)*:
  **return** *(title$_i$, url$_i$, snippet$_i$)$_{i=1}^k$*;

**Function** *visit(u, q)*:
  $D \leftarrow \text{scrape}(u)$;
  $D \leftarrow \text{split}(D) = \{d_i\}_{i=1}^m$;
  if $q = \emptyset$ then **return** $d' \leftarrow d_1$;
  else $d' \leftarrow \arg\max_{d_i \in D} \text{F1}(d_i, q)$;
  **return** $d'$;

**Result:** Task output $y$

Turn $t \leftarrow 1$;
Context $C \leftarrow \{x\}$;
$\mathcal{T} \leftarrow \{\text{search}\}$;
**while** $t < T$ **do**
  $o_t \leftarrow \theta(C; \mathcal{T})$;               `/* LLM may only call the search tool */`
  **switch** $o_t$ **do**
    **case** *search* **do**
      $R \leftarrow \text{search}(o_t)$;               `/* Perform search */`
      $l \leftarrow \text{length}(C)$;
      $D \leftarrow \{c_i\}_{i=l-5}^l$;
      **for** $(t_i, u_i, s_i) \in R$ **do**
        $D \leftarrow D \cup \text{visit}(u_i, s_i)$;      `/* Visit every search result */`
      $C \leftarrow C \cup \{o_t, \theta(D; \text{summarize})\}$;
    **case** *Final Answer* **do**
      **return** $o_t$;
  $t \leftarrow t + 1$;
**return** $\theta(C; \textit{final answer})$;

---

1. **Search Agent**: an agent that will search the internet to answer a question.

2. **Visualizer**: given the path to a downloaded image, it will call an LLM to answer questions about the image.

3. **Text Inspector**: given the path to a downloaded text file, it will call an LLM to answer questions about the text.

The search agent has access to the following tools:

1. **Google Search**: a search engine that will search the internet to answer a question. This tool uses Serper API in the backend.

2. **Visit Tool**: visit a URL and render the page in HTML as in a browser.

3. **Page Up**: navigate the current page by scrolling up.

4. **Page Down**: navigate the current page by scrolling down.

5. **Finder Tool**: find a text in the current page.

6. **Find Next**: find the next occurrence of the text in the current page.

7. **Archive Search**: search the archives for information.

8. **Text Inspector**: given the path to a downloaded text file, it will call an LLM to answer questions about the text.

Detailed descriptions of each tool can be found in the original implementation[10].

---

[10]https://github.com/huggingface/smolagents/blob/main/src/smolagents/default_tools.py

---

**Algorithm 3:** HuggingFace Open Deep Research

---

**Data:** Task input $x$, LLM $\theta$, maximum number of turns for search and main agents $T_s$ and $T_m$, respectively, and planning interval $p$

web_tools ←
{Search, Visit, Page Up, Page Down, Finder, Find Next, Archive Search, Text Inspector};
main_tools ← {search_agent, Visualize, Text Inspector};
**Function** *plan(q, c)*:
   /* Prompt the LLM to generate a plan                     */
   **return** $\theta(q, c; plan)$;
**Function** *search_agent(q)*:
   $P \leftarrow \text{plan}(q, \emptyset)$;
   $C \leftarrow \{q, P\}$;
   $t \leftarrow 1$;
   **while** $t < T_s$ **do**
      **if** $t \mod p = 0$ **then**
         $P \leftarrow \text{plan}(q, C)$;
         $C \leftarrow C \cup \{P\}$;
      $o_t \leftarrow \theta(C; \text{web\_tools})$;
      **if** $type(o_t) = final\_answer$ **then**
         **return** $o_t$;
      /* do the tool call, see A.2 for tool details          */
      $C \leftarrow C \cup \{o_t, \text{tool}(o_t)\}$;
      $t \leftarrow t + 1$;
   **return** $\theta(C; final\ answer)$;
**Result:** Task output $y$
Turn $t \leftarrow 1$;
$P \leftarrow \text{plan}(x, \emptyset)$;
Context $C \leftarrow \{x, P\}$;
/* the main agent plans and calls the search agent          */
**while** $t < T_m$ **do**
   **if** $t \mod p = 0$ **then**
      $P \leftarrow \text{plan}(x, C)$;
      $C \leftarrow C \cup \{P\}$;
   $o_t \leftarrow \theta(C; \text{main\_tools})$;
   **if** $type(o_t) = final\_answer$ **then**
      **return** $o_t$;
   $C \leftarrow C \cup \{o_t, \text{tool}(o_t)\}$;
   $t \leftarrow t + 1$;
**return** $\theta(C; final\ answer)$;

---

### A.3 TRAJECTORY-LEVEL ANALYSIS DEFINITIONS

In this subsection, we describe how we annotate each trajectory with the failure modes. For LLM-as-a-judge approaches, we use o3-2025-04-16 as the judge model. In each of the following LLM-as-a-judge approaches, we use the same judge model, and force the model to generate its response in a json format for easy parsing. We find that existing frontier LLMs are powerful enough to reliably check for simple yes/no questions and output them in a json format.

**Confirmation bias.** Confirmation bias occurs when the system finds a potential candidate that is incorrect in its search process, and subsequently spends the majority of its search budget on the same candidate without considering other options, leading to a lack of exploration in the search space. To detect this, we first collect all the search queries that the system has made and then use an LLM to check if the search queries overly focus on a single wrong candidate. The judge model is given access to the groundtruth answer and the search queries, so it's able to determine if the search queries are

---

**Algorithm 4:** GPT-Researcher

---

**Data:** Task input $x$, LLM $\theta$, research depth $D$, research breadth $B$, summary interval $n$

**Function** *search(q)*:

    **return** *(title$_i$, url$_i$, snippet$_i$)$_{i=1}^{k}$*;

**Function** *visit(u, q)*:

    $D \leftarrow \text{scrape}(u)$;

    $D \leftarrow \text{split}(D) = \{d_i\}_{i=1}^{m}$;

    if $q = \emptyset$ then **return** $d' \leftarrow d_1$;

    else $d' \leftarrow \arg\max_{d_i \in D} \text{F1}(d_i, q)$;

    **return** $d'$;

**Function** *plan(q)*:

    /* Prompt the LLM to generate a list of queries          */

    $R \leftarrow \text{search}(q)$;

    **return** $\theta(x, R; plan)$;

**Function** *conduct_research(q)*:

    /* Conduct research on one query by generating subqueries and

        retrieve and scrape            */

    $Q \leftarrow \text{plan}(q)$;

    $R \leftarrow \emptyset$;

    **for** $q_i \in Q$ **do**

        **for** $t_i, u_i, s_i \in search(q_i)$ **do**

            $r_i \leftarrow \text{visit}(u_i, s_i)$;

            $R \leftarrow R \cup r_i$;

    **return** $\theta(x, R; process)$;

**Function** *deep_research(q, d)*:

    /* Recursively plan and conduct research           */

    $Q \leftarrow \text{plan}(q)$;

    $R \leftarrow \emptyset$;

    **for** $q_i \in Q$ **do**

        $r_i \leftarrow \text{conduct\_research}(q_i)$;

        /* Prompt the LLM to generate takeaways and follow up

            questions            */

        $q'_i \leftarrow \theta(r_i; process)$;

        **if** $d < D$ **then**

            $R \leftarrow R \cup \text{deep\_research}(q'_i, d+1)$;

    **return** $R$;

**Result:** Task output $y$

Turn $t \leftarrow 1$;

Context $C \leftarrow \{x\}$;

$P \leftarrow \text{plan}(x)$;

$R \leftarrow \text{deep\_research}(P, 1)$;

**return** $\theta(R; write\ report)$;

---

relevant to the groundtruth answer and the similarities between different search queries. We consider a trajectory to have confirmation bias if a majority of the search queries are similar to each other, and focuses on a single wrong candidate. The prompt used for confirmation bias detection is shown in Table 4.

**Unfocused search.** Unfocused search occurs when the system generates overly generic search queries that are not useful for narrowing down the search space—the system cannot make any progress towards finding useful information. To detect this, we first collect all the search queries that the system has made and then use an LLM to check if the search queries are generic and not useful for narrowing down the search space. We consider a trajectory to have unfocused search if a majority

| **Prompt for Confirmation Bias Detection** |
|---|
| You are a helpful assistant that can analyze the trajectory of an information-seeking agent. You are given a question-answer pair and the search history of an agent that tried to answer the question. You should analyze the search history and determine if the agent spends more than half of the tool calls searching for the same incorrect answer. That is, the agent continues searching for the same topic even though it's not the correct answer to the question, and spends half or more of its tool calls on these searches. Output your final conclusion with your reasoning and a single word: 'yes' if the agent spends more than half of its tool calls on the same incorrect answer or 'no' if the agent does not. **Reasoning:** explain what the agent did, and if it did or did not focus its searches on a wrong answer. **Conclusion:** "yes" or "no". **Search queries:** `<search-queries>` **Question:** `<question>` **Correct Answer:** `<correct-answer>` |

Table 4: System prompt used for detecting confirmation bias in agent trajectories

| **Prompt for Unfocused Search Detection** |
|---|
| You are a helpful assistant that can analyze the trajectory of an information-seeking agent. You are given a question-answer pair and the search history of an agent that tried to answer the question. You should analyze the search history and determine if the search queries do not help the agent narrow down the search space. Consider the following cases: 1. The agent searches for information relevant to the question and answer, but it's not specific enough to yield helpful results. 2. The agent searches for queries that are not sufficiently relevant or specific to the question and answer, which does not narrow down the search space enough. 3. The agent explores the search space with diverse queries but does not use enough tool calls to properly narrow down the search space by either eliminating wrong answers or verifying the correct answer. All of these cases are considered to be unfocused search. You should consider the whole trajectory of the agent, and not just some of the tool calls—only consider the trajectory to be unfocused if more than half of the searches are unfocused. Output your final conclusion with your reasoning and a single word: 'yes' if the searches are unfocused or 'no' if the searches are focused enough. **Reasoning:** explain what the agent did, and if it did or did not use tool calls to properly narrow down the search space. **Conclusion:** "yes" or "no". **Search queries:** `<search-queries>` **Question:** `<question>` **Correct Answer:** `<correct-answer>` |

Table 5: System prompt used for detecting unfocused search in agent trajectories

of the search queries are overly generic and not useful for narrowing down the search space. The prompt used for unfocused search detection is shown in Table 5.

**Inefficient tool usage.** Inefficient tool usage occurs when the system does not discover new information with its tool calls, and is therefore wasting its tool budget. Specifically, we use URLs as a proxy for the information discovered by the system—a tool call that only return URLs seen in previous search results is considered as a waste of tool budget. We use a simple heuristic for this analysis—iterate over all search calls made in the trajectory and keep track of seen URLs. Then, we report the percentage of search calls that only return URLs seen in previous search results.

**Answer ignored.** Answer ignored occurs when the system encounters the correct answer in its search process, but does not use it to answer the question. One possible explanation is that the system is distracted by other noisy information in its context, preventing it from correctly identifying the groundtruth. We employ a simple approach for this analysis—we check if the groundtruth answer is

---

**Prompt for Groundtruth Ignored Detection**

You are a helpful assistant that can analyze the trajectory of an information-seeking agent. You are given a question-answer pair and a list of webpages. You should analyze the web contents and determine if it contains the correct answer. The correct answer is considered to be found if there are some context in the search results that is either a direct or near-exact match to the correct answer. Output your final conclusion with your reasoning and a single word: 'yes' if the content contains the correct answer or 'no' if the content does not contain the correct answer.

**Reasoning:** explain if the web content contains the correct answer.

**Conclusion:** "yes" or "no".

`<tool-responses>`

**Question:** `<question>`

**Correct Answer:** `<correct-answer>`

---

Table 6: System prompt used for detecting groundtruth ignored in agent trajectories

---

**Prompt for Giving Up Detection**

You are a helpful assistant that can analyze the final output of an information-seeking agent. You are to check if the agent decides that it cannot find the correct answer. For example, if the explanation states that it cannot find enough relevant information to answer the question, or if the response is simply empty or "I don't know", then the agent did not attempt to answer the question. Output your final conclusion with a single word "yes" if the agent decides it did not find enough information to answer the question or "no" otherwise.

**Conclusion:** "yes" or "no".

**Final output:** `<final-output>`

---

Table 7: System prompt used for detecting giving up in agent trajectories

present in any of the tool responses. We employ a LLM judge to enable fuzzy matching between the groundtruth answer and the tool responses. The prompt used for answer ignored detection is shown in Table 6. We iterate over all tool calls and use this check to determine if any tool responses contain the groundtruth answer. We terminate the iteration if we find a tool response that contains the groundtruth answer, and report the percentage trajectories where at least one tool response contains the groundtruth answer.

**Abstention.** Abstention occurs when the system does not attempt to answer the question due to the lack of information in its context. Existing LLMs can often refuse to answer the question if it is not confident in answering the question, but this behavior is not desirable for search agents that could leverage additional tool calls to find the necessary information. We use a simple LLM judge to check if the system attempted to answer the question. The prompt used for giving up detection is shown in Table 7.

**Hallucination.** Hallucination occurs when the system generates information that is not supported by the information it has discovered in its search process. In agentic search systems, it is not desirable to hallucinate information, as it could result in incorrect and misleading answers and thus affect the trustworthiness of the system. Inspired by previous works(Rashkin et al., 2023; Bohnet et al., 2022; Gao et al., 2023), we check if the system hallucinates information by first decomposing the model's explanation into a set of atomic claims. Then, we iterate through all the tool responses from the search process and check if the tool responses support all the claims. As long as one tool response support a claim, we consider the system to not have hallucinated that claim. In the end, we report the average percentage of unsupported claims across trajectories. The prompt used for decomposing the model's explanation into a set of atomic claims is shown in Table 8, and the prompt used for hallucination detection is shown in Table 9. These prompts are derived from previous works that show LLMs can reliably decompose texts into a set of atomic claims and check if claims are supported by a piece of text—they also achieve high agreement with human judges (Gao et al., 2023; Kamoi et al., 2023; Yen et al., 2025).

**Prompt for Decomposing Explanation into Atomic Claims**

Read the given explanation and generate a list of atomic claims that are supported by the explanation. Atomic claims that are basic facts that cannot be further broken down. Generate at most 10 claims for the explanation.

Use the following as an example:

```
Explanation:  Searching UFCStats for featherweight bouts
where the loser landed 14 of 83 significant strikes (16.87 %)
and went 0-for-4 on takedowns returns the fight Myles Jury
vs. Ricky Glenn at UFC 219: Cyborg vs Holm (30 Dec 2017).
• Ricky Glenn (nickname "The Gladiator"|a synonym
for swordsman)
was the loser: sig. strikes 14/83 (16.87 %), takedowns 0/4.
• Both fighters (Jury 29, Glenn 28) were under 35 and
are American.
• The referee was John McCarthy, whose first event for
the UFC was in 1994.
Thus, the MMA event is UFC 219: Cyborg vs Holm.

Exact Answer: UFC 219: Cyborg vs Holm

Confidence: 75%
```

**Atomic Claims**:
- Ricky Glenn was the loser
- Ricky Glenn was nicknamed "The Gladiator"
- The sig. strike rate of Ricky Glenn was 14/83 (16.87- The takedown rate of Ricky Glenn was 0/4
- Jury was age 29
- Glenn was age 28
- Jury is American
- Glenn is American
- The referee was John McCarthy
- John McCarthy's first event for the UFC was in 1994
Output the atomic claims in the form of a json list.

Table 8: System prompt used for decomposing the model's explanation into a set of atomic claims

**Prompt for Hallucination Detection**

You are a helpful assistant that can analyze the trajectory of an information-seeking agent. You are given a list of webpages and a list of claims made by the agent. You should analyze the web contents to determine if each claim is supported by the web content. A claim is supported by the web content if its factual information is mostly supported by the web content, and is not contradicted by the web content. Output your final conclusion with a list of claims that are supported by the web content. Output the list in the form of a json list, and you only need to write the index of the supported claims in the list and nothing else.
**Webpages:** `<webpages>`
**Atomic Claims:** `<atomic-claims>`

Table 9: System prompt used for detecting hallucination in agent trajectories

## A.4 SLIM DETAILS AND ABLATIONS

We show an example of a SLIM trajectory in Figure 8. A sketch of the framework is also shown in Alg. 5. Furthermore, we ablate our design choices along the following dimensions:

- **Summarization frequency:** Instead of summarizing the trajectory every $n = 50$ turns, we summarize every $n = 25$ turns.

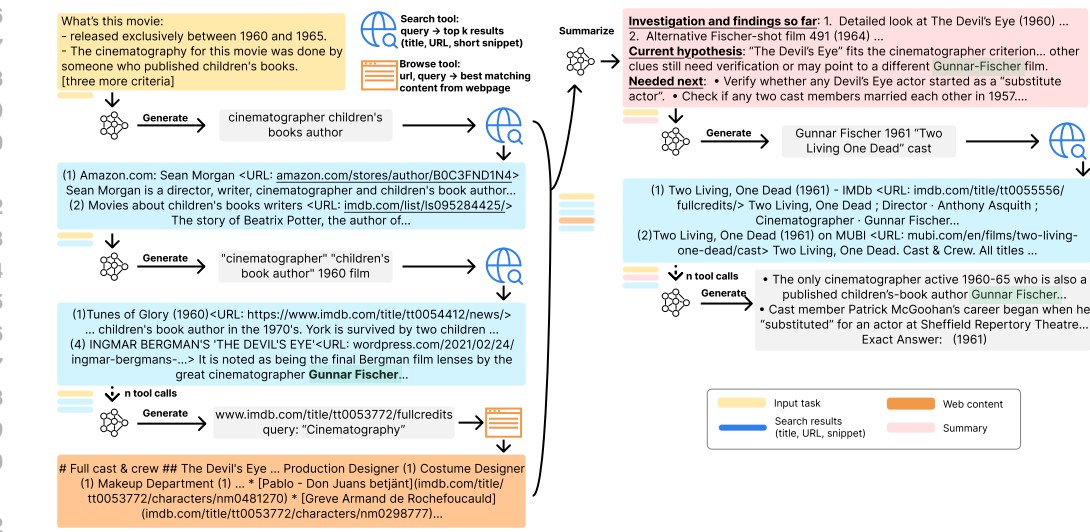

Figure 8: An example of a SLIM trajectory.

---

**Algorithm 5:** SLIM

**Data:** Task input $x$, LLM $\theta$, maximum number of turns $T$, summary interval $n$

**Function** *search(q)*:
 **return** $(title_i, url_i, snippet_i)_{i=1}^{k}$;

**Function** *browse(u, q)*:
 $D \leftarrow \text{scrape}(u)$;
 $D \leftarrow \text{split}(D) = \{d_i\}_{i=1}^{m}$;
 **if** $q = \emptyset$ **then return** $d' \leftarrow d_1$;
 **else** $d' \leftarrow \arg\max_{d_i \in D} \text{ROUGE-L}(d_i, q)$;
 **return** $d'$;

**Result:** Task output $y$

Turn $t \leftarrow 1$;
Context $C \leftarrow \{x\}$;
$\mathcal{T} \leftarrow \{\text{search}, \text{browse}\}$;
**while** $t < T$ **do**
 **if** $t \mod n = 0$ **then**
  $C \leftarrow \theta(C; \text{summarize})$ ;     /* Summarize every $n$ turns */
 $o_t \leftarrow \theta(C; \mathcal{T})$;
 **switch** $o_t$ **do**
  **case** *search* **do**
   $q_t \leftarrow o_t$;
   $C \leftarrow C \cup \{o_t, \text{search}(q_t)\}$;
  **case** *browse* **do**
   $u_t, s_t \leftarrow o_t$;
   $C \leftarrow C \cup \{o_t, \text{browse}(u_t, s_t)\}$;
  **case** *Final Answer* **do**
   **return** $o_t$;
 $t \leftarrow t + 1$;
**return** $\theta(C; \text{final answer})$;

---

- **Summarization trigger:** Instead of summarizing the trajectory every $n$ turns, we summarize the trajectory when the input length exceeds a threshold $\tau = \{32768, 65536\}$ tokens.

- **Search tool:** We vary the number of top search results $k = \{10, 20\}$.

- **Browse tool:** We vary the maximum length of the scraped content $L = \{3000, 10000, 20000\}$ characters. We also ablate the chunking and scoring strategy. By default, we chunk by natural paragraphs (splitting at newlines) and use ROUGE-L as the similarity metric. We also try using BM25 (Robertson & Zaragoza, 2009) as the similarity metric and splitting the content into chunks of 100 words (splitting at any whitespace).

For these ablations, we use o4-mini as the base model due to its cheaper cost and test on a smaller subset of 50 samples for each dataset. The results are shown in Table 10.

Table 10: Ablation results with o4-mini as the base model. The number of tokens is shown in 10,000s. The cost is shown in US dollars. We ablate design choices in the summarization module, chunking strategy, and search and browse tool. For all settings, we set the tool budget to 100. The default setting summarizes every $n = 50$ turns, chunks by newline, use ROUGE-L as the similarity metric, and search returns the top $k = 10$ search results while browsing returns at most $L = 10,000$ characters. These experiments use a smaller subset of 100 samples for each dataset, so they are not directly comparable to the main results. Each experiment is run with three random seeds and the results are the mean and standard deviation.

| | | BrowseComp | | | | HLE | | | |
|---|---|---|---|---|---|---|---|---|---|
| | | Score (↑) | Tokens (↓) | Tools (↓) | Cost (↓) | Score (↑) | Tokens (↓) | Tools (↓) | Cost (↓) |
| SLIM | Default | 40.67±5.86 | 118.27±5.93 | 54.02±2.43 | 1.33±0.07 | 17.33±3.06 | 11.39±1.34 | 7.61±0.46 | 0.13±0.01 |
| **Summarization Module** | | | | | | | | | |
| $n = 25$ | | 30.33±4.51 | 57.64±4.87 | 35.47±1.04 | 0.65±0.05 | 21.67±4.04 | 8.53±1.68 | 6.09±0.96 | 0.1±0.02 |
| Summarize at 32K tokens | | 29.67±2.89 | 46±3.23 | 32.7±0.79 | 0.53±0.04 | 17.67±6.51 | 9.22±1.34 | 6.62±0.69 | 0.11±0.02 |
| Summarize at 64K tokens | | 42.67±2.08 | 126.2±5.69 | 57.23±2.14 | 1.42±0.06 | 19.67±4.51 | 11.67±0.74 | 7.83±0.34 | 0.13±0.01 |
| **Chunking** | | | | | | | | | |
| Split newline, BM25 | | 37.67±3.51 | 121.38±8.56 | 55.3±2.01 | 1.37±0.1 | 21.33±4.04 | 12.21±1.14 | 8.33±0.31 | 0.14±0.01 |
| Split words, ROUGE | | 39.33±5.03 | 113.55±2.88 | 52.97±1.39 | 1.28±0.03 | 19.33±5.69 | 11.69±1.05 | 7.91±0.44 | 0.13±0.01 |
| Split words, BM25 | | 40.67±2.52 | 121.39±3.33 | 55.95±1.18 | 1.37±0.04 | 20.33±4.16 | 10.4±0.66 | 7.32±0.53 | 0.12±0.01 |
| **Search and Browse** | | | | | | | | | |
| No visit | | 34.33±2.89 | 111.53±5.1 | 63.47±2.03 | 1.26±0.06 | 15.33±1.15 | 14.35±1.68 | 10.42±0.9 | 0.16±0.02 |
| No query in visit | | 37.33±1.15 | 187.63±9.54 | 66.82±2.34 | 2.1±0.11 | 20.33±2.08 | 14.55±0.75 | 8.9±0.62 | 0.17±0.01 |
| $k = 10, L = 3,000$ | | 42±6.24 | 111.59±12.51 | 52.77±3.95 | 1.26±0.14 | 21.33±1.53 | 11.65±1.2 | 7.75±0.07 | 0.13±0.01 |
| $k = 10, L = 20,000$ | | 38.67±1.53 | 117.35±7.79 | 54.5±1.53 | 1.32±0.09 | 20.67±0.58 | 12.19±1.24 | 7.84±0.57 | 0.14±0.01 |

## A.5 EXPERIMENTAL DETAILS

We use o3, o4-mini, and Claude-4-Sonnet as our base models. To calculate the cost, we use the prices listed in Table 11, which are obtained from respective websites `https://platform.openai.com/docs/models/o3`, `https://platform.openai.com/docs/models/o4-mini`, `https://claude.com/pricing#api`, `https://www.firecrawl.dev/pricing`.

For all models, we use a temperature of $1.0$ and a maximum output token of $32,768$. For o3 and o4-mini, we always use the default reasoning effort of "medium" and for Claude-4-Sonnet, we set the maximum number of thinking tokens to $30,000$.

To calculate the token cost, we take a weighted sum of the token usage across all LLM calls: non-cached input tokens plus 4 times the total output tokens, and multiply the results by price per token. We exclude cached tokens from the calculation because in practice, long-horizon systems are expected to have a large amount of cached tokens and system implementation that takes advantage of caching. Then, for the total cost, we add in the number of search API and scrape URL operations, multiplied by their respective prices. For the number of tool calls, we count the number of times the search API and scrape operations, the two atomic tool operations, are called.

We also include the results of other trained systems in Table 12. For OpenAI Deep Research (DR), the HLE number from the original blog post[11] and the BrowseComp number is from the BrowseComp paper (Wei et al., 2025). For Grok-4, the HLE number is from the original Grok 4 blog post [12] and the BrowseComp number is from the Grok 4 Fast blog post [13]. The WebResearcher (WebR) numbers

---

[11] `https://openai.com/index/introducing-deep-research/`
[12] `https://x.ai/news/grok-4`
[13] `https://x.ai/news/grok-4-fast`

are from the original paper (Qiao et al., 2025), where we show the results of the main WebResearcher-30B-A3B model; we exclude the heavy version since it uses multiple samples and aggregate the results. The WebThinker (WebT) numbers are from the original paper (Li et al., 2025c), where we show the results of the main WebThinker-32B model. They did not evaluate on BrowseComp, so we only report the HLE number.

Table 11: Pricing for different components. Numbers are obtained from respective websites.

|  | Cost |
| --- | --- |
| o3 | $2.0 / M token |
| o4-mini | $1.1 / M token |
| Claude-4-Sonnet | $3.0 / M token |
| Google search | $0.5 / K query |
| Scrape URL | $0.83 / K query |

## A.6 ADDITIONAL RESULTS

**Main Results.** We show the results of SLIM with o3 as the base model over three random seeds in Table 13. Here we also provide the concrete results for SLIM with different base models—o4-mini is shown in Table 14, and Claude-4-Sonnet is shown in Table 15.

Table 12: Main results with o3 as the base model. All results are macro-averaged across test instances. The number of tokens is shown in 10,000s. The cost is shown in US dollars. $T$ denotes the tool budget. For reference only, † marks deep research systems that underwent task-specific training. Numbers are from the original reports (OpenAI, 2025; xAI, 2025; Qiao et al., 2025; Li et al., 2025c), and are not directly comparable due to different subsets of test instances used.

|  |  | BrowseComp | | | | HLE | | | |
| --- | --- | --- | --- | --- | --- | --- | --- | --- | --- |
|  | $T$ | Score ($\uparrow$) | Tokens ($\downarrow$) | Tools ($\downarrow$) | Cost ($\downarrow$) | Score ($\uparrow$) | Tokens ($\downarrow$) | Tools ($\downarrow$) | Cost ($\downarrow$) |
| o3 | 0 | 17.0 | 3.8 | 0.0 | 0.08 | 18.3 | 2.7 | 0.0 | 0.05 |
| REACT | 1 | 4.3 | 3.6 | 1.0 | 0.07 | 16.0 | 4.6 | 0.6 | 0.09 |
|  | 5 | 6.7 | 6.6 | 2.2 | 0.13 | 19.7 | 5.8 | 1.1 | 0.12 |
|  | 10 | 7.0 | 8.0 | 2.8 | 0.16 | 21.3 | 7.0 | 1.2 | 0.14 |
| SEARCH-O1 | 1 | 18.0 | 3.8 | 9.5 | 0.08 | 20.0 | 3.3 | 5.2 | 0.07 |
|  | 5 | 24.0 | 8.0 | 46.9 | 0.20 | 20.7 | 5.4 | 18.7 | 0.12 |
|  | 10 | 31.0 | 13.7 | 89.8 | 0.35 | 26.3 | 6.6 | 23.9 | 0.15 |
|  | 25 | 40.0 | 27.8 | 183.2 | 0.70 | 25.0 | 10.9 | 44.2 | 0.25 |
|  | 50 | 48.3 | 51.5 | 306.2 | 1.27 | 27.0 | 12.6 | 49.8 | 0.29 |
|  | 100 | 55.7 | 93.3 | 456.7 | 2.23 | 27.0 | 14.5 | 52.2 | 0.33 |
| HF-ODR | 20 | 20.0 | 24.1 | 8.4 | 0.49 | 17.7 | 6.4 | 1.7 | 0.13 |
| GPT-R | - | 10.7 | 5.8 | 69.5 | 0.17 | 16.0 | 6.4 | 85.6 | 0.20 |
| SLIM | 10 | 17.7 | 2.7 | 8.7 | 0.06 | 22.7 | 4.2 | 3.8 | 0.09 |
|  | 25 | 32.7 | 9.0 | 20.7 | 0.19 | **31.3** | 7.7 | 6.9 | 0.16 |
|  | 50 | 45.0 | 25.0 | 36.0 | 0.52 | 31.0 | 13.6 | 9.7 | 0.28 |
|  | 100 | 53.3 | 44.1 | 57.4 | 0.91 | **31.3** | 18.4 | 11.6 | 0.37 |
|  | 150 | **56.0** | 59.8 | 75.9 | 1.24 | 30.7 | 17.9 | 12.0 | 0.37 |
| OpenAI DR† | - | 51.5 | - | - | - | 26.6 | - | - | - |
| Grok-4† | - | 43.0 | - | - | - | 38.6 | - | - | - |
| WebR-30B† | - | 37.3 | - | - | - | 28.8 | - | - | - |
| WebT-32B† | - | 15.8 | - | - | - | - | - | - | - |

**REACT Ablations.** We vary the number of search results $k$ and the maximum length of the scraped content $L$ for REACT to see the effect of search tool design choices, as shown in Table 16. We found that overall there aren't significant differences in the HLE results, but using fewer search results $k = 5$ than the default $k = 10$ leads to a 2.7 points improvement in the BrowseComp results. This is likely due to the fact that search results lower in the ranking are often noisy and irrelevant to the question, and using fewer but more relevant search results leads to a more focused search process. Furthermore, fewer search results means less context is added to the LLM, preventing it from hitting

Table 13: Statistical significance analysis with o3 as the base model. We run with three random seeds for each experiment and report the mean and standard deviation.

| | | BrowseComp | | | | HLE | | | |
|---|---|---|---|---|---|---|---|---|---|
| | | Score (↑) | Tokens (↓) | Tools (↓) | Cost (↓) | Score (↑) | Tokens (↓) | Tools (↓) | Cost (↓) |
| o3 | - | 17.22±1.02 | 3.87±0.12 | 0±0 | 0.08±0 | 19.56±1.07 | 2.63±0.04 | 0±0 | 0.05±0 |
| Search-o1 | 50 | 49.33±1.2 | **49.98±1.5** | 298.9±6.84 | 1.24±0.04 | 26.78±0.69 | **13.05±0.52** | 50.96±1.17 | **0.3±0.01** |
| SLIM | 150 | **53±1.2** | 54.77±5.23 | **50.84±0.44** | **1.12±0.1** | **32.11±1.84** | 16.44±1.15 | **10.3±0.98** | 0.33±0.02 |

Table 14: Main results with o4-mini as the base model. All results are macro-averaged across test instances. The number of tokens is shown in 10,000s. The cost is shown in US dollars. $T$ denotes the maximum number of turns in each trajectory.

| | | BrowseComp | | | | HLE | | | |
|---|---|---|---|---|---|---|---|---|---|
| | $T$ | Score (↑) | Tokens (↓) | Tools (↓) | Cost (↓) | Score (↑) | Tokens (↓) | Tools (↓) | Cost (↓) |
| o4-mini | - | 5.0 | 5.1 | 0.0 | 0.06 | 15.0 | 2.2 | 0.0 | 0.02 |
| REACT | 1 | 1.3 | 4.6 | 1.0 | 0.05 | 17.0 | 4.0 | 0.5 | 0.04 |
| | 5 | 3.0 | 7.7 | 2.1 | 0.09 | 15.3 | 4.6 | 0.7 | 0.05 |
| | 10 | 2.3 | 7.4 | 2.3 | 0.08 | 15.3 | 4.9 | 0.8 | 0.05 |
| SEARCH-O1 | 1 | 6.3 | 6.2 | 10.0 | 0.08 | 13.0 | 2.6 | 3.5 | 0.03 |
| | 5 | 11.3 | 13.8 | 49.7 | 0.19 | 23.3 | 4.0 | 11.9 | 0.05 |
| | 10 | 17.3 | 22.6 | 93.9 | 0.32 | 17.0 | 4.6 | 15.6 | 0.06 |
| | 25 | 25.0 | 45.4 | 207.7 | 0.66 | 22.3 | 5.5 | 22.5 | 0.08 |
| | 50 | 28.7 | 76.1 | 351.5 | 1.12 | 19.3 | 7.3 | 26.3 | 0.10 |
| | 100 | 36.0 | 124.4 | 546.7 | 1.80 | 21.3 | 6.6 | 25.8 | 0.09 |
| HF-ODR | 20 | 15.0 | 38.9 | 15.4 | 0.44 | 16.3 | 8.3 | 3.9 | 0.09 |
| GPT-R | - | 4.0 | 8.5 | 82.5 | 0.16 | 11.3 | 9.7 | 100.8 | 0.19 |
| SLIM | 10 | 14.0 | 5.7 | 8.8 | 0.07 | 21.0 | 3.6 | 3.1 | 0.04 |
| | 25 | 24.3 | 24.0 | 23.2 | 0.28 | 23.7 | 7.2 | 5.9 | 0.08 |
| | 50 | 31.0 | 73.7 | 40.1 | 0.83 | 25.7 | 10.0 | 7.0 | 0.11 |
| | 100 | 34.0 | 92.9 | 45.2 | 1.05 | **26.7** | 12.2 | 7.7 | 0.14 |
| | 150 | **37.0** | 107.8 | 49.5 | 1.22 | 24.7 | 14.4 | 8.6 | 0.16 |

Table 15: Main results with Claude-4-Sonnet as the base model. All results are macro-averaged across test instances. The number of tokens is shown in 10,000s. The cost is shown in US dollars. $T$ denotes the maximum number of turns in each trajectory.

| | | BrowseComp | | | | HLE | | | |
|---|---|---|---|---|---|---|---|---|---|
| | $T$ | Score (↑) | Tokens (↓) | Tools (↓) | Cost (↓) | Score (↑) | Tokens (↓) | Tools (↓) | Cost (↓) |
| Claude-4-Sonnet | - | 1.0 | 1.9 | 0.0 | 0.06 | 6.3 | 3.9 | 0.0 | 0.12 |
| REACT | 1 | 0.3 | 0.0 | 0.0 | 0.00 | 8.3 | 0.0 | 0.0 | 0.00 |
| | 5 | 0.3 | 0.0 | 0.0 | 0.00 | 8.3 | 0.0 | 0.0 | 0.00 |
| | 10 | 0.3 | 0.0 | 0.0 | 0.00 | 8.0 | 0.0 | 0.0 | 0.00 |
| SEARCH-O1 | 1 | 2.0 | 1.5 | 9.0 | 0.05 | 10.0 | 2.9 | 10.0 | 0.09 |
| | 5 | 3.7 | 6.0 | 44.1 | 0.21 | 11.7 | 5.3 | 29.5 | 0.18 |
| | 10 | 7.0 | 10.7 | 79.5 | 0.38 | 16.0 | 6.3 | 35.6 | 0.22 |
| | 25 | 8.0 | 20.1 | 149.9 | 0.72 | 13.0 | 6.8 | 41.1 | 0.24 |
| | 50 | 10.0 | 22.9 | 170.3 | 0.82 | 12.7 | 7.0 | 40.7 | 0.24 |
| | 100 | 10.0 | 19.4 | 148.3 | 0.70 | 12.3 | 6.4 | 38.5 | 0.22 |
| HF-ODR | 20 | 6.7 | 98.8 | 30.4 | 2.98 | 17.3 | 105.0 | 26.5 | 3.16 |
| GPT-R | - | 2.3 | 7.9 | 106.5 | 0.32 | 8.0 | 6.9 | 94.9 | 0.28 |
| SLIM | 10 | 2.7 | 2.8 | 8.9 | 0.09 | 10.3 | 2.5 | 6.9 | 0.08 |
| | 25 | 9.7 | 5.1 | 21.6 | 0.17 | 15.0 | 2.8 | 10.2 | 0.09 |
| | 50 | 10.0 | 5.0 | 27.1 | 0.16 | 17.3 | 3.0 | 9.9 | 0.10 |
| | 100 | **10.7** | 4.8 | 28.1 | 0.16 | 14.0 | 2.9 | 10.5 | 0.09 |
| | 150 | 10.0 | 5.2 | 30.7 | 0.17 | **16.7** | 3.1 | 11.1 | 0.10 |

the context window limit as much. This is evident in more token and tool usage. However, we use $k = 10$ for the main experiments to stay consistent with the other baselines.

Table 16: REACT ablations with o3 as the base model, and the maximum number of turns is $T = 10$. We vary the number of search results $k$ and the maximum length of the scraped content $L$.

| | Parameters | | | BrowseComp | | | | HLE | | | |
|---|---|---|---|---|---|---|---|---|---|---|---|
| | $T$ | $k$ | $L$ | Score (↑) | Tokens (↓) | Tools (↓) | Cost (↓) | Score (↑) | Tokens (↓) | Tools (↓) | Cost (↓) |
| REACT | 10 | 10 | 10k | 7.0 | 8.0 | 2.8 | 0.16 | 21.3 | 7.0 | 1.2 | 0.14 |
| REACT | 10 | 5 | 10k | 9.7 | 10.6 | 4.1 | 0.21 | 21.7 | 7.0 | 1.7 | 0.14 |
| REACT | 10 | 10 | 3k | 5.0 | 8.7 | 2.8 | 0.18 | 22.7 | 6.5 | 1.2 | 0.13 |
| REACT | 10 | 5 | 3k | 8.3 | 10.7 | 4.1 | 0.22 | 21.3 | 6.7 | 1.7 | 0.13 |

## A.7 OPEN-WEIGHT MODELS

In this subsection, we show the results of SLIM with open-weight models GPT-OSS-120B (OpenAI et al., 2025) and Tongyi-DeepResearch, an RL-trained model for deep research (Team, 2025). We compare against the SEARCH-O1 setting with similar total cost. The results are shown in Table 17 and Table 18. We observe similar improvement with our framework SLIM over competitive baselines. Controlling for cost, SLIM achieves significant improvements on BrowseComp.

Table 17: Results with GPT-OSS-120B as the base model. We compare against the SEARCH-O1 setting with similar total cost.

| | | BrowseComp | | | | HLE | | | |
|---|---|---|---|---|---|---|---|---|---|
| | | Score (↑) | Tokens (↓) | Tools (↓) | Cost (↓) | Score (↑) | Tokens (↓) | Tools (↓) | Cost (↓) |
| GPT-OSS-120B | - | 2.67 | 1.35 | 0.00 | 0.00 | 7.00 | 1.07 | 0.00 | 0.00 |
| SEARCH-O1 | 10 | 12.67 | 8.28 | 79.28 | 0.08 | 11.67 | 2.29 | 12.56 | **0.01** |
| SLIM | 150 | **15.33** | **3.37** | **22.28** | **0.02** | **20.33** | **1.72** | **5.32** | **0.01** |

Table 18: Results with Tongyi-DeepResearch-30B as the base model. We compare against the SEARCH-O1 setting with similar total cost.

| | | BrowseComp | | | | HLE | | | |
|---|---|---|---|---|---|---|---|---|---|
| | | Score (↑) | Tokens (↓) | Tools (↓) | Cost (↓) | Score (↑) | Tokens (↓) | Tools (↓) | Cost (↓) |
| Tongyi-DeepResearch-30B | - | 2.33 | 7.07 | 0.00 | 0.03 | 11.00 | 5.58 | 0.00 | 0.02 |
| SEARCH-O1 | 10 | 14.33 | 13.80 | 70.25 | 0.11 | **20.00** | 12.24 | 44.39 | 0.08 |
| SLIM | 150 | **19.67** | **12.35** | **61.59** | **0.08** | 19.67 | **10.10** | **23.12** | **0.05** |

## A.8 ADDITIONAL ANALYSIS

In this subsection, we provide additional analysis—we extend the initial outcome-based analysis to SLIM, and show the trajectory-level analysis on the more comprehensive baselines.

In Table 19, we show the trajectory-level analysis where we report the failure modes as a percentage of trajectories that ends with an incorrect answer. The trends are consistent with the analysis in the main text, but we find that SLIM can often find the correct answer across its long trajectories—over 69% of the incorrect trajectories encounters the correct answer, but the model is not able to identify and use it to answer the question. This could be attributed to the fact that modern LLMs still struggle at long-context settings where it may need to reason over many sources. We leave these improvements to future work.

Table 19: For correct, we report the percentage of trajectories across all samples. For each trajectory-level failure mode, we report the percentage of trajectories that ends with an incorrect answer. For hallucination only, we report the percentage of hallucinations for samples that ends with an incorrect answer and do not abstain.

| Framework | Turn Budget | Correct | Confirm Bias | Unfocused Search | Inefficient Search | Abstention | Answer Ignored | Hallucinate |
|---|---|---|---|---|---|---|---|---|
| REACT | 10 | 7.0 | 10.0 | 47.3 | 4.2 | 1.1 | 0.7 | 56.7 |
| SEARCH-O1 | 50 | 48.3 | 18.1 | 65.2 | 14.0 | 8.4 | 50.3 | 46.8 |
| HF-ODR | 20 | 20.0 | 8.6 | 75.5 | 56.5 | 41.6 | 2.1 | 96.2 |
| SLIM | 150 | 56.0 | 22.0 | 77.3 | 17.2 | 62.9 | 69.7 | 19.0 |

