# OpenReview forum: "Lost in the Maze: Overcoming Context Limitations in Long-Horizon Information-Seeking"
_ICLR.cc/2026/Conference — Submitted to ICLR 2026_

### Official Review · Reviewer_Gggp · 2025-10-20

**Soundness:** 3
**Presentation:** 3
**Contribution:** 2
**Rating:** 4
**Confidence:** 4

**Summary:**

This paper proposes a lightweight agent framework for long-horizon information-seeking. By limiting search depth, avoiding unnecessary webpage browsing, and performing periodic summarization, the method aims to reduce tool usage and context length while maintaining answer quality.

**Strengths:**

- The paper focuses on an important problem in long-horizon agent research—context bloat and noise accumulation during multi-step search.

- The framework is simple, easy to reproduce, and may have practical applicability due to its low engineering overhead.

- Experiments cover multiple benchmarks, and results are generally stable.

**Weaknesses:**

- Missing comparisons with the most relevant literature.
Many existing agent systems (e.g., InfoGENT, memory-based agents) already include context compression and memory management mechanisms that target the same issue. Without comparing to these, the claimed contribution and improvement are not well supported.

- Metric design raises fairness concerns.
Counting tool calls while excluding webpage crawling, whereas prior agents include it, makes the comparison less meaningful. It is unclear whether token-cost metrics also exclude webpage reading. The advantage may come from accounting choices rather than methodological benefits.

- Limited insight and unclear mechanism.
The paper does not clearly explain why the framework improves performance—is it better planning, cleaner evidence selection, or simply shorter text? Improvements seem small and potentially within sampling noise, without offering deeper understanding or design principles.

**Questions:**

See above.

---

> ### Author Response · Authors · 2025-11-24
>
> We thank the reviewer for the helpful feedback, and we are encouraged that the reviewer found our design simple and practical. We address the concerns below.
>
> | W1: Missing comparisons with the most relevant literature
>
> All of the frameworks we compare against have some form of context/memory management: Search-o1 conducts summarization on the retrieved results, HF-ODR leverages sub-agents and explicit planning steps, and GPT-researcher uses parallel agents and planning steps. We chose the most popular open-source agentic search and deep research systems for comparison. These systems also span both simple single-agent and complex multi-agent systems, which we believe serve as a representative and fair group of baselines for the paper. The key difference in our approach is using a simple trajectory-level summarization that effectively reduces the length and noise in the agent context.
> While other baselines, such as InfoGent, may be insightful to include in the experiments, they are expensive to adapt while they are conceptually similar to some of our baseslines already.
>
> | W2: Metric design raises fairness concerns. Counting tool calls while excluding webpage crawling, whereas prior agents include it, makes the comparison less meaningful.
>
> As stated in Section 5 and Appendix A.5, we **do** include webpage crawling in the tool call counter, which we referred to as the scraping operation. This means a fair comparison with previous agents and should address your concern. Furthermore, the cost metric includes all tool calls (search and page crawl/browse) and LLM usage, which is an accurate reflection of the cost that practitioners would incur when they run these systems.
>
> | W3: Limited insight and unclear mechanism. Improvements seem small and potentially within sampling noise
>
> Please see the general response for a better understanding of our error taxonomy and automated pipeline for a fine-grained analysis of the failure modes of existing systems. We also ran our experiments with 3 different random seeds and found significant improvements beyond sampling noise.

---

> > ### Comment · Reviewer_Gggp · 2025-11-27
> >
> > Thank you very much for your careful work and detailed responses. I have updated my score accordingly

---

### Official Review · Reviewer_eygJ · 2025-10-30

**Soundness:** 3
**Presentation:** 4
**Contribution:** 2
**Rating:** 4
**Confidence:** 3

**Summary:**

The paper presents an approach for long-horizon search agents and evaluates it against a benchmark. The results are promising and show reduced computational effort with increased effectiveness.

**Strengths:**

- The paper presents interesting results that follow the recent trend of agentic frameworks for information seeking and deep research.

. The authors use recent valid benchmark tasks and data

- Some insights on why current approaches may fail are provided

**Weaknesses:**

- The main weakness of the paper is that it is merely a combination of existing tools that can overcome the context limitation of current models

- There is limited methodological contribution in the task alignment or agent models themselves

- Some of the results are not correctly highlighted for best performance

**Questions:**

- Lack of statistical evidence on whether the results are significantly different from baselines

- Lack of insights on why existing systems fail. If I understood correctly this is mainly because of context limitations and the advantage of the proposed approach is more practical than scientific

---

> ### Author Response · Authors · 2025-11-24
>
> We thank the reviewer for the helpful feedback, and we are encouraged that the reviewer found our results promising and used recent datasets.
>
> | W1:  a combination of existing tools that can overcome the context limitation of current models. | W2: limited methodological contribution in the task alignment or agent models themselves
>
> An important highlight of SLIM over previous frameworks—ReAct, Search-o1, HF-ODR, and GPT-Researcher—is that it achieves notably more efficient and effective results using a simpler design, which we deem as a useful contribution to the community. The research community, including the listed works, has sought to overcome the context limitations through various approaches that involve increasingly more complex architectures: Search-o1 conducts summarization on the retrieved results, HF-ODR leverages sub-agents, and GPT-researcher uses parallel agents. While these approaches show progress, we argue that they also demonstrate significant limitations in efficiency and are harder to analyze and improve. Instead, we show via strong empirical results that simple and principled designs actually result in the strongest performance, which we hope can serve as a strong message to the community to seek for simpler and more scalable approaches.
> Furthermore, a core contribution of our work is the fine-grained trajectory-level error taxonomy and analysis pipeline. This leads to a better understanding of how and why current systems still fail on difficult agentic search tasks, informing future developments of these systems.
>
> | W3: some results are not correctly highlighted for best performance
>
> If the reviewer is referring to the presentation of the main results in Table 3, we only highlight the best score, as it’s uninformative to highlight the lowest token usage, tool usage, and cost, because the base model achieves the lowest across these metrics. However, the line plots (Figure 1, 5, and 6) are better illustrations of the Pareto frontiers.
>
> | Q1:  Lack of statistical evidence on whether the results are significantly different from baselines
>
> Please see the general response for our main results with multiple runs, which should address the concern here
>
> | Q2: Lack of insights on why existing systems fail
>
> Please see the general response for a better understanding of our error taxonomy and automated pipeline for a fine-grained analysis of the failure modes of existing systems.

---

> > ### Comment · Reviewer_eygJ · 2025-11-25
> > **Acknowledgement of response**
> >
> > Thank you for your response. I acknowledge it, but will maintain my position.

---

### Official Review · Reviewer_yUMY · 2025-11-01

**Soundness:** 3
**Presentation:** 3
**Contribution:** 3
**Rating:** 4
**Confidence:** 3

**Summary:**

This paper targets long-horizon information-seeking systems and argues that context explosion is the main bottleneck preventing existing agent frameworks from scaling. The authors propose SLIM, a lightweight method separating search and browse operations and periodically summarizing interaction history. Experiments on BrowseComp and HLE show comparable or better performance to several open-source baselines with lower tool usage.

**Strengths:**

1. Identifies a practical but overlooked bottleneck in agent-based research systems.
2. Simple yet effective design; strong empirical support.
3. Thorough evaluation across multiple models and benchmarks.
4. Trajectory-level error taxonomy adds clarity and interpretability to agent failures.

**Weaknesses:**

1. Core components are heuristic and not deeply analyzed (e.g., why summarize every n steps?).
2. Lack of rigorous ablations and theoretical justification for the design decisions.
3. Results are promising but not clearly superior in all cases; advantages are modest on some metrics.
4. Does not explore whether SLIM benefits RL-trained or task-specialized agents.

**Questions:**

1. How sensitive is SLIM to summarization interval n? Could adaptive summarization outperform fixed cadence?
2. Would a memory-selective mechanism (retain only high-value facts) be more efficient than compressing the full trajectory?
3. How robust is SLIM to noisy or domain-specific search engines?

---

> ### Author Response · Authors · 2025-11-24
>
> We thank the reviewer for the helpful feedback, and we are encouraged that the reviewer found our design simple yet effective while achieving strong empirical results. We address the concerns below:
>
> | W1: Core components are heuristic and not deeply analyzed
> | W2: Lack of rigorous ablations and theoretical justification for the design decisions
> | Q1: Sensitivity of summarization interval n
>
> Please see our general response where we summarize our rigorous ablation studies. We ablate across several axes (i.e., the summarization module and the browse tool) with multiple runs. We further test different summarization intervals. These results should address the raised concerns here.
>
> | W3: Results are promising but not clearly superior in all cases; advantages are modest on some metrics
>
> Across all models we tested in the main paper (o3, o4-mini, and Claude-4-Sonnet), SLIM achieves significant improvement across both benchmarks when controlling for the total cost. Please see the general response for our main results with multiple runs, which further validate the significance of SLIM’s advantage. Furthermore, our additional experiments on GPT-OSS also found similar improvements.
> Could the reviewer please clarify which metrics show more modest advantages between SLIM and the compared approaches?
>
> | W4: whether SLIM benefits RL-trained or task-specialized agents
>
> Please see the general response for experiments on Tongyi-DeepResearch, a competitive RL-trained agent for deep research tasks. We found similar improvements with SLIM.
>
> | Q2: Would a memory-selective mechanism (retain only high-value facts) be more efficient than compressing the full trajectory?
>
> We find that existing summarization models already do something similar to memory-selective mechanisms through qualitative analysis. In the example trajectory we show in Figure 8, the model summarizes the trajectory into several bullet points, such as “Investigation and findings so far”, “Current hypothesis”, and “Needed next”. The resulting summary is similar to many memory-selective mechanisms that only retain relevant facts to the current query.
> Thus, we find that allowing the model to compress the full trajectory naturally filters out irrelevant information while achieving simplicity and avoiding over-prompt-engineering.
>
> | Q3: How robust is SLIM to noisy or domain-specific search engines?
>
> The quality of search engines is critical to agentic search systems, and SLIM offers the model greater flexibility in interleaving the search calls and browse calls, which would likely allow it to perform better than frameworks that do not enable this flexibility, like Search-o1.
> Being more robust to potentially noisy search tools is exactly one of SLIM’s advantages. As we show in the paper, compared to other frameworks, SLIM can more easily scale up many search calls while keeping only the relevant information via its better context management mechanism, overcoming noisy content returned by the search engine. On the other hand, frameworks like Search-o1 and GPT-Researcher are forced to browse every page the search engine returns.
> Similar logic applies to domain-specific search engines as well—greater flexibility should allow the model to adapt to new domains better. We also show this by running the experiments on HealthBench, a long-form generation task where the model must respond to issues as if they were a physician; please see the general response for this experiment.

---

> ### Comment · Reviewer_yUMY · 2025-11-28
> **Response to Authors**
>
> I appreciate the authors’ detailed and constructive rebuttal. The additional experiments, expanded ablations, and responses to the raised concerns help clarify several aspects of the work. In particular, the broader ablation studies on the summarization module, summarization intervals, and browse tool design address my earlier concerns regarding heuristic choices (W1/W2), and the new results across multiple random seeds strengthen the empirical claims. The clarification regarding performance under cost-controlled settings (W3) is also helpful.
>
> Overall, I find the rebuttal reasonable and the added experiments convincing. Strengthening the clarity and visibility of these results in the camera-ready version, especially for summarization sensitivity, memory selectivity, and robustness claims, would make the contribution more complete and persuasive.

---

### Official Review · Reviewer_ge5u · 2025-11-01

**Soundness:** 3
**Presentation:** 2
**Contribution:** 3
**Rating:** 6
**Confidence:** 3

**Summary:**

This paper addresses context management challenges in deep research systems. The authors propose SLIM (Simple Lightweight Information Management), a framework that separates retrieval into distinct search and browse tools while periodically summarizing trajectories to maintain concise context. Evaluated on part of BrowseComp and Human Last Exam benchmarks, SLIM outperforms the open-source baselines in accuracy and the number of tool calls. The paper also introduces an automated trajectory analysis pipeline with an error taxonomy to characterize failure modes, revealing that SLIM exhibits fewer unfocused searches and less hallucination than existing systems.​

**Strengths:**

1. SLIM shows strong gains on challenging benchmarks. It outperforms all open-source baselines by a substantial margin (e.g., +8 points on BrowseComp). Importantly, it achieves these gains efficiently: SLIM requires only ~15–25% of the tool calls that competitors need to reach similar turn counts. The results hold across multiple base LLMs (o3, o4-mini, Claude-4-Sonnet) and both task domains, suggesting the high generalization of the proposed approach.

2. Beyond the high performance of SLIM, the paper contributes a trajectory error taxonomy and an automatic trajectory analysis pipeline. By categorizing failure modes (e.g. unfocused search, confirmation bias, hallucination) and annotating agent trajectories, the authors gain deeper insight. They show SLIM’s advantage is largely due to dramatically reduced hallucination rates compared to others, a valuable finding. This analysis is itself a novel contribution and is well-integrated into the paper.

3. The paper is well-written and structured. The authors also release source code to reproduce the experiments. Overall, the presentation is clear and the method is easy to follow.

**Weaknesses:**

1. The evaluation scope is limited:

1.1. The experiments focus on two specific benchmarks (BrowseComp and HLE). While both are challenging, it is unclear how SLIM would perform on other kinds of tasks (e.g., open-domain QA, shorter queries, or multimodal search). Both BrowseComp and HLE focus on factoid question answering with short, verifiable answers. This limits our understanding of how SLIM performs on​ open-ended research questions requiring synthesis across sources, multi-hop reasoning requiring complex information integration, and tasks where the answer requires aggregation rather than finding a single fact. The paper acknowledges this implicitly by noting answers are "short and straightforward, resulting in reliable evaluation", but this also means the evaluation doesn't cover the full spectrum of deep research tasks mentioned in the motivation.​

1.2. Only part (300 samples) from each dataset are used, making it difficult to compare the results with metric values from previous studies. Though it is reasonable to conduct most experiments on such a subset, it is still important to report the results of the best configuration of the proposed technique on complete sets, and present its comparison with SOTA results obtained by other researchers.

1.3. The sample size (300) can be not enough for drawing strong conclusions about a method's effectiveness. Moreover, the paper provides no statistical significance testing, confidence intervals, or error bars. Without knowing the variance in performance, it's unclear whether the reported improvements are statistically significant. For example, the 7-point improvement on BrowseComp could be within noise if variance is high across different runs or random seeds.​

2.  The choice of baselines (REACT, SEARCH-O1, HF-ODR, GPT-Researcher) should be explained better. There exist a lot of open-source implementations of deep research systems (see, e.g., survey https://arxiv.org/abs/2506.12594 or Universal Deep Research from NVidia), so there more thorough discussion and, potentially, experimental study, are important.

3. The most significant methodological flaw is the absence of a proper ablation study. SLIM has 3 key components (separated search/browse, periodic summarization), but it's unclear how much each component contributes to the overall performance and efficiency gains depending on the problem and on the base model. Does the summarization module provide a significant boost over just using the separated tools? Is the `browse` tool's selective fetching more important than the `search` tool's conciseness? Without this analysis, it's difficult to attribute the success to the full design versus a single key component. How does performance vary with different summarization strategies, the change in search engine (Google) and web scraper (crawl4ai)?

4. The summarization module, while crucial, could introduce its own errors (e.g., loss of critical details). The paper does not analyze what kind of information is lost during summarization and whether this ever leads to failure. A qualitative analysis of summarization errors would strengthen the claims.

5. The error taxonomy is insightful, but it is manually defined from BrowseComp examples. It’s unclear how well these categories apply to other domains or how reliable the heuristic/LLM-based annotator is. Some of the failure modes (e.g. “answer ignored”) are a bit ambiguous.

6. The paper ignores existing papers that analyze the fine-grained failures of multi-agent systems, e.g., TRAIL (https://arxiv.org/abs/2505.08638) or MAST taxonomy (https://arxiv.org/abs/2503.13657). To highlight the contribution of the proposed trajectory error taxonomy, it is necessary to compare it with similar ideas.

7. The paper is focused on open-source baselines in deep research, but all used LLMs are proprietary. Though I understand that they may be much more accurate compared to open-source LLMs, experiments with at least one open LLM and other open tools (search engine, etc.) will significantly increase the reproducibility.

**Questions:**

1. What are the limitations of the proposed deep research? In what scenarios would it work worse compared to existing systems?

2. The browse tool fetches the “most relevant section” of a page via a similarity function. How do you split the document into separate sections? Did you compare this against simply fetching the whole page (or multiple snippets)?

3. Have you tried SLIM on shorter-horizon tasks (e.g. conventional question-answering) or tasks without web search? Would the search/browse framework still be beneficial in those scenarios?

4. The automated pipeline tags trajectories with errors. Can you clarify how much manual effort is needed for this? How confident are you that “hallucination” is correctly detected by the system?

---

> ### Author Response · Authors · 2025-11-24
> **Rebuttal 1/3**
>
> We thank the reviewer for the detailed comments and useful feedback. We are encouraged that the reviewer found our framework strong, the error taxonomy novel and insightful, and the paper well-written. We address the weaknesses and questions below:
>
> | W1.1: Focus on two specific benchmarks: HLE and BrowseComp
>
> We focus on HLE and BrowseComp due to their popularity in the research community; nearly all recent works evaluate on these two benchmarks [1][2][3]. Although the scope of this paper is on long-horizon agentic search tasks, we agree that it may be useful to evaluate on other kinds of tasks that better understand how our framework may translate to other settings.
> Thus, we evaluate on a subset of HealthBench, which is a long-form generation task where the agent must respond to users’ medical questions like a physician [4]. This dataset was collected and annotated by real-world physicians and grades the long-form generation using rubric items. For these experiments, we evaluate SLIM (with budget = 150), Search-o1 (with budget = 50, most comparable setting to ours in terms of cost), and other baselines in the paper. Due to cost, we evaluate with o4-mini as the base model and randomly sample 300 instances from the HealthBench Hard set.
>
> |                  | overall_score | axis:completeness | axis:accuracy | axis:communication_quality | axis:instruction_following |
> |-----------------------------|---------------|-------------------|---------------|----------------------------|----------------------------|
> | num rubric items            | -             | 1425              | 1033          | 198                        | 172                        |
> | o4-mini                     |         18.19 |             19.27 |         35.08 |                      61.46 |                      49.66 |
> | Search-o1-o4-mini-50   |         21.61 |              21.4 |     **38.79** |                  **64.79** |                  **56.04** |
> | HF-ODR-o4-mini              |             0 |                 0 |         16.73 |                      52.44 |                      35.88 |
> | GPT-researcher-o4-mini      |             0 |                 0 |         29.34 |                      16.42 |                      35.12 |
> | SLIM-o4-mini-150 |     **22.99** |         **23.48** |         37.34 |                      60.65 |                      51.69 |
>
> In the table, we show the overall score, the axes that each response is rated on, and the number of rubric items in each axis. SLIM achieves the highest overall scores compared to other baselines. Note the physician-written rubric highly prioritizes completeness of the response (e.g., covering different treatments and considerations of a medical condition), which has the most overall number of rubric items, and SLIM outperforms all other baselines on this axis.
> While SLIM sees comparable results and some underperformance on the other axes, we find it promising that our frameworks overall work best for a long-generation task as well.
>
> [1] Liu et al., 2025. WebExplorer: Explore and Evolve for Training Long-Horizon Web Agents
> [2]Li et al., 2025. WebSailor-V2: Bridging the Chasm to Proprietary Agents via Synthetic Data and Scalable Reinforcement Learning
> [3] Qiao et al., 2025. WebResearcher: Unleashing unbounded reasoning capability in Long-Horizon Agents
> [4] Arora et al., 2025. HealthBench: Evaluating Large Language Models Towards Improved Human Health
>
> | W1.2: report the results of the best configuration of the proposed technique on complete sets
>
> We report the results of SLIM-o4-mini on the full set of both datasets.
>
> |                  | BrowseComp | HLE   |
> |------------------|------------|-------|
> | o4-mini          |       4.42 | 15.11 |
> | SLIM-o4-mini-150 |      29.07 | 23.54 |
>
> Due to time and cost constraints, we do not report the performance of the other baselines—it would have taken over a week or more to run Search-o1, HF-ODR, or GPT-Researcher on one dataset. However, as the reviewer stated, the results can still be useful for comparing with other works.
>
> | W1.3: The sample size (300) can be not enough for drawing strong conclusions about a method's effectiveness
>
> Please see the general response for results with multiple seeds, where we show that the noise across runs is relatively small and SLIM improves significantly over baselines.
>
> | W2: Choice of baselines
>
> Thank you for the comment, we will improve the related works section and section 3.2 with additional discussions. We chose the most popular open-source agentic search and deep research systems for comparison. These systems also span both simple single-agent and complex multi-agent systems, which we believe serve as a representative and fair group of baselines for the paper. Due to the high cost and long runtime of agentic systems, we only evaluate the representative baselines.

---

> ### Author Response · Authors · 2025-11-24
> **Rebuttal 2/3**
>
> | W3:  absence of a proper ablation study
>
> Please see the general response for a rigorous study where we test several different summarization approaches, as well as various approaches to the search and browse tools.
>
> | W4: A qualitative analysis of summarization errors
>
> In the example trajectory we show in Figure 8, the model summarizes the trajectory into several bullet points, such as “Investigation and findings so far”, “Current hypothesis”, and “Needed next”.
> From our qualitative analysis, we observe that existing models do a good job at summarizing the main points of the trajectories, such as the queries and findings so far, and we do not observe any major issues. This is likely due to the fact that existing models have already been trained extensively for a common task like summarization. We will include more discussions and examples in the final revision.
>
> | W5: unclear how well these categories apply to other domains or how reliable the heuristic/LLM-based annotator is. Some of the failure modes (e.g. “answer ignored”) are a bit ambiguous.
>
> The purpose of our error taxonomy is to better understand agentic information-seeking tasks, where the agent exhaustively searches over long trajectories with a short-form ground truth answer. Thus, these categories are specifically designed for such tasks by examining the search queries, results, and final outputs. We believe that such a taxonomy is useful for the community to better study this problem.
>
> First, our heuristic approach is deterministic and reliable in nature. For example, if a search query does not yield any new unseen URL in its search result, then it does not provide any new information, which is then categorized as an inefficient search. Such an error can be reliably checked with a simple heuristic approach.
> Second, using LLM annotators as judges for certain evaluation tasks is well-studied in previous literature, including for question-answering and hallucination checking [5][6][7][8][9]. In our qualitative analysis of the LLM annotators, we found that existing models are reliable checkers for simple tasks, aligned with findings from previous works.
> Third, the short oracle answer of the tasks further reduces the ambiguity for the annotator LLM significantly.
> For example, in the category “answer ignored”, we provide the judge model with the ground truth answer as well as the web contents presented to the agent, and ask the model whether the answer can be found in the content. This essentially checks for exact match overlap between the content and the answer, where if the answer is found in the content, then it means that the agent ignored it by outputting the wrong answer.
> Given the popularity of using judge models to check fuzzy matching, we believe that this is a reliable metric to measure.
>
> [5] Zheng et al., NeurIPS 2023. Judging LLM-as-a-Judge with MT-Bench and Chatbot Arena
> [6] Gu et al., 2025. A Survey on LLM-as-a-Judge
> [7] Yen et al., ICLR 2025. HELMET: How to Evaluate Long-context Language Models Effectively and Thoroughly
> [8] He et al., NeurIPS 2025. Precise Information Control in Long-Form Text Generation
> [9] Leng et al., DataBricks Blog 2023. Best Practices for LLM Evaluation of RAG Applications
>
>
> | W6: paper ignores existing papers that analyze the fine-grained failures of multi-agent systems
>
> We thank the reviewer for providing specific works to compare our analysis framework with. The key difference between our error taxonomy and previous works like TRAIL and MAST is that our focus is on the agentic search task. Specifically, we designed our taxonomy to analyze different aspects and stages in which the search agent may fail in a long trajectory, such as issuing repetitive and inefficient search, failing to recover from a wrong search path, or ignoring the final answer when generating the final output. Our analysis dives deeper into the tool usage and tool results of the search agents.
> We design our automated analysis pipeline to conduct fine-grained analysis across a search trajectory, while previous works study more general multi-agent interaction. The two approaches, general and specific, are complementary to each other in gaining a better understanding of agentic systems.
>
> | W7: only proprietary LLMs
>
> Please see the general review for our experiments on open-weight models, such as GPT-OSS-120B and Tongyi-DeepResearch-30B, where we also observe superior performance with SLIM over competitive baselines similar to proprietary models.

---

> ### Author Response · Authors · 2025-11-24
> **Rebuttal 3/3**
>
> | Q1: What are the limitations of the proposed deep research? In what scenarios would it work worse compared to existing systems?
>
> The proposed deep research system is naturally limited by the ability of the base model. Although we show that it works well for powerful base models, such as o3, Claude, and some open-source models like GPT-OSS and Tongyi-DeepResearch, the performance may be limited by the base model’s ability to scale to long trajectories and correctly use tools. For weaker base models, perhaps more prompt-heavy approaches (e.g., repeatedly asking the model to refine tool calls) may be better through more intervention in the overall system. However, with the improvement of recent base models, these may not be a concern.
>
> | Q2: The browse tool fetches the “most relevant section” of a page via a similarity function. How do you split the document into separate sections? Did you compare this against simply fetching the whole page (or multiple snippets)?
>
> We split the documents into separate sections via natural paragraphs (splitting at newlines). However, in our ablations, we also test splitting the documents by words and using different similarity functions (e.g., BM25). We also compare it against displaying longer content when browsing; we don’t simply present the entire document due to very long web pages and PDFs that would easily make the context window overflow. Please see the general response for these experiments.
>
> | Q3: Have you tried SLIM on shorter-horizon tasks (e.g. conventional question-answering) or tasks without web search? Would the search/browse framework still be beneficial in those scenarios?
>
> We do not study SLIM on shorter-horizon tasks or tasks without web search, such as conventional QA, because the focus of our work is on long-horizon information-seeking tasks, which challenge even the frontier models with their planning and context management capabilities . For conventional QA tasks, it’s unlikely to observe a significant difference from the baselines as those tasks typically only require few search calls and rarely run into context limitations.
>
> | Q4: The automated pipeline tags trajectories with errors. Can you clarify how much manual effort is needed for this? How confident are you that “hallucination” is correctly detected by the system?
>
> Using the pipeline to tag trajectories with our error taxonomy does not require any manual effort—our supplementary materials contain the code for doing this (which will be open-sourced). We are confident that hallucinations are correctly detected by the system due to the number of previous studies that confirm the reliability of LLM judges at doing this task [7][8][10]. These works confirm the reliability of LLM judges by showing high human-model agreement.
>
> [7] Yen et al., ICLR 2025. HELMET: How to Evaluate Long-context Language Models Effectively and Thoroughly
> [8] He et al., NeurIPS 2025. Precise Information Control in Long-Form Text Generation
> [10] Kamoi et al., EMNLP 2023. Wice: Real-world entailment for claims in wikipedia

---

> ### Comment · Reviewer_ge5u · 2025-11-26
> **Thanks for additional experiments**
>
> Thanks for your additional experiments and detailed answers! In general, I believe it is possible to modify the paper by merging additional results with current content, add comparison with SOTA results on complete benchmarks known from literature.
> In addition, could you kindly explain the difference in metrics for a complete set of BrowseComp and your subsample of 300 examples? SLIM-o4-mini-150 on the full dataset, not a subset, achieved a quality of 29.07 for BrowseComp, while the subsample had a quality of 37.0, if I'm not mistaken (Table 12 in the Appendix). These are significant differences, meaning the subset isn't representative of the entire dataset. The second HLE (HumanLastExam) is better: 24.7 for the subset, and 23.54 for the full dataset.

---

> > ### Author Response · Authors · 2025-11-26
> >
> > Thank you for the fast response, we will update the paper with the additional results and comparisons. For the differences between the subset and the full set on BrowseComp, it's likely that, as you said, the subset is not representative of the full dataset. Thus, two approaches should only be compared on the same set of evaluation samples, where we can indeed see improvements over baselines on the subset over 3 random seeds. As for HLE, the differences between the subset (24.7) and the full set (23.54) is small enough that it could be explained by noise—we saw that the std for HLE scores are around 1-2.

---

### Author Response · Authors · 2025-11-24
**General Response Rebuttal (1/2)**

We thank all reviewers for their insightful comments and useful feedback. We are encouraged that the reviewers found our framework to be a simple yet effective approach to tackle a challenging and motivated problem in agentic search systems. We address some common concerns below:

| Statistical significance

We reran our main experiments using three random seeds and show the standard deviation across them.

|           |        | BrowseComp |               |                |              | HLE            |                |               |              |
|-----------|--------|------------|---------------|----------------|--------------|----------------|----------------|---------------|--------------|
|           | Budget | score      | tokens        | tools          | cost         | score          | tokens         | tools         | cost         |
| o3        | -      | 17.22±1.02 | 3.87±0.12     | 0±0            | 0.08±0       | 19.56±1.07     | 2.63±0.04      | 0±0           | 0.05±0       |
| Search-o1 | 50     | 49.33±1.2  | **49.98±1.5** | 298.9±6.84     | 1.24±0.04    | 26.78±0.69     | **13.05±0.52** | 50.96±1.17    | **0.3±0.01** |
| SLIM      | 150    | **53±1.2** | 54.77±5.23    | **50.84±0.44** | **1.12±0.1** | **32.11±1.84** | 16.44±1.15     | **10.3±0.98** | 0.33±0.02    |

We observe that our framework SLIM achieves significant improvement over the next best baseline with comparable costs and 5-6x fewer tool calls. In general, the noise across different runs is relatively low, even when the evaluation size is 300 instances, suggesting that it’s sufficient for comparing approaches. We will extend the statistical analysis to other baselines in the revision.

| SLIM with open-source models and RL-trained models

We conduct our experiments with the [GPT-OSS-120B](https://huggingface.co/openai/gpt-oss-120b) model with the base model, our framework, and the Search-o1 setting with comparable total cost. For the cost calculation, we use the token cost from TogetherAI, a model serving platform, and all other settings are identical to our main experiments.

|              |        | BrowseComp |          |           |          | HLE       |          |          |          |
|--------------|--------|------------|----------|-----------|----------|-----------|----------|----------|----------|
|              | budget | score      | tokens   | tools     | cost     | score     | tokens   | tools    | cost     |
| GPT-OSS-120B | -      | 2.67       | 1.35     | 0.00      | 0.00     | 7.00      | 1.07     | 0.00     | 0.00     |
| Search-o1    | 10     | 12.67      | 8.28     | 79.28     | 0.08     | 11.67     | 2.29     | 12.56    | **0.01** |
| SLIM         | 150    | **15.33**  | **3.37** | **22.28** | **0.02** | **20.33** | **1.72** | **5.32** | **0.01** |

We find that with GPT-OSS, our framework outperforms competitive baselines, while achieving lower costs and fewer tool calls, suggesting the effectiveness of our framework across open and closed models. We also

Similarly, we also use [Tongyi-DeepResearch-30B](https://huggingface.co/Alibaba-NLP/Tongyi-DeepResearch-30B-A3B) as the model for our experiments. This model was specifically RL-trained for deep research. We compare SLIM against the most competitive open-source baseline, Search-o1, with similar costs, for which we calculate the token cost from OpenRouter, where the model is served through API calls.

|                         |     | BrowseComp |           |           |          | HLE       |           |           |          |
|-------------------------|-----|------------|-----------|-----------|----------|-----------|-----------|-----------|----------|
|                         |     | score      | tokens    | tools     | cost     | score     | tokens    | tools     | cost     |
| Tongyi-DeepResearch-30b |     | 2.33       | 7.07      | 0.00      | 0.03     | 11.00     | 5.58      | 0.00      | 0.02     |
| Search-o1               | 10  | 14.33      | 13.80     | 70.25     | 0.11     | **20.00** | 12.24     | 44.39     | 0.08     |
| SLIM                    | 150 | **19.67**  | **12.35** | **61.59** | **0.08** | 19.67     | **10.10** | **23.12** | **0.05** |

We find that SLIM achieves superior performance on BrowseComp and similar performance on HLE while using much fewer tools and lower cost. Thus, our simple design is effective while lowering cost compared to other open frameworks.

---

### Author Response · Authors · 2025-11-24
**General Response Rebuttal (2/2)**

| Rigorous ablations

We ablate our design choices for the summarization module, search tool, and browse tool by running experiments on another random subset of 100 instances from the dataset. We show the mean and standard deviation across three random seeds below. Due to cost concerns, we run with o4-mini as the base model and set the maximum tool budget to 100.

|                                         | BrowseComp |              |            |           | HLE        |            |           |           |
|-----------------------------------------|------------|--------------|------------|-----------|------------|------------|-----------|-----------|
|                                         | score      | tokens       | tools      | cost      | score      | tokens     | tools     | cost      |
| SLIM-o4-mini (n=50, newline, ROUGE, l=10k) | 40.67±5.86 | 118.27±5.93  | 54.02±2.43 | 1.33±0.07 | 17.33±3.06 | 11.39±1.34 | 7.61±0.46 | 0.13±0.01 |
| Summarization module                    |            |              |            |           |            |            |           |           |
| n=25                                    | 30.33±4.51 | 57.64±4.87   | 35.47±1.04 | 0.65±0.05 | 21.67±4.04 | 8.53±1.68  | 6.09±0.96 | 0.1±0.02  |
| t=32k                                   | 29.67±2.89 | 46±3.23      | 32.7±0.79  | 0.53±0.04 | 17.67±6.51 | 9.22±1.34  | 6.62±0.69 | 0.11±0.02 |
| t=64k                                   | 42.67±2.08 | 126.2±5.69   | 57.23±2.14 | 1.42±0.06 | 19.67±4.51 | 11.67±0.74 | 7.83±0.34 | 0.13±0.01 |
| Browse tool                             |            |              |            |           |            |            |           |           |
| No browse tool                               | 34.33±2.89 | 111.53±5.1   | 63.47±2.03 | 1.26±0.06 | 15.33±1.15 | 14.35±1.68 | 10.42±0.9 | 0.16±0.02 |
| No query in browse                      | 37.33±1.15 | 187.63±9.54  | 66.82±2.34 | 2.1±0.11  | 20.33±2.08 | 14.55±0.75 | 8.9±0.62  | 0.17±0.01 |
| newline, BM25                           | 37.67±3.51 | 121.38±8.56  | 55.3±2.01  | 1.37±0.1  | 21.33±4.04 | 12.21±1.14 | 8.33±0.31 | 0.14±0.01 |
| words, ROUGE                            | 39.33±5.03 | 113.55±2.88  | 52.97±1.39 | 1.28±0.03 | 19.33±5.69 | 11.69±1.05 | 7.91±0.44 | 0.13±0.01 |
| words, BM25                             | 40.67±2.52 | 121.39±3.33  | 55.95±1.18 | 1.37±0.04 | 20.33±4.16 | 10.4±0.66  | 7.32±0.53 | 0.12±0.01 |
|  l=3k                              | 42±6.24    | 111.59±12.51 | 52.77±3.95 | 1.26±0.14 | 21.33±1.53 | 11.65±1.2  | 7.75±0.07 | 0.13±0.01 |
| l=20k                             | 38.67±1.53 | 117.35±7.79  | 54.5±1.53  | 1.32±0.09 | 20.67±0.58 | 12.19±1.24 | 7.84±0.57 | 0.14±0.01 |

For the summarization module, we test with different summarization intervals: every 25 turns, or every 32/64K tokens. We found that summarizing every 50 turns is better than or comparable with the other approaches.

For the browse tool, we tested not having the browse tool at all and no query parameter when browsing, which resulted in much lower BrowseComp scores. We also tested different chunking strategies, such as splitting the documents into chunks of 100 words instead by new lines, and similarity scores, such as BM25 instead of ROUGE. Finally, we also tested showing different lengths of content in the browse tool—by default, we show 10k characters, but we also tried showing 3k and 20k characters in browse. In general, we found that our default setting had similar or better BrowseComp scores, but observed some lower HLE scores that are still within the standard deviation.

Thus, the most critical part of the framework appears to be having long enough content before summarization (i.e., longer summarization intervals in terms of turns or tokens) and having the browse tool, while other design choices (i.e., the similarity function and chunking strategy) do not have a great impact on the results.

| lack of insight into why SLIM improves over previous systems

A core contribution of our paper is the error taxonomy of the failure modes in existing systems, as outlined in Section 6. Our extensive qualitative analysis results in an error taxonomy that covers many aspects of agentic search failure models—we analyzed the search queries, results, and the final outputs. These analyses provide fine-grained insights into each framework’s trajectories. For example, SLIM tends to hallucinate significantly less than previous systems, likely due to the fact that there is less irrelevant noise in the context to distract the model.

Furthermore, through our detailed ablations above, we find that the use of the Browse tool greatly improves performance on long-horizon search tasks.

---

### Author Response · Authors · 2025-12-03
**Discussion Period Summary**

We want to thank all reviewers for their helpful feedback and engagement during the discussion period. Overall, we are encouraged that the reviewers found our work (1) evaluation settings thorough and empricically strong across models  (ge5u, yUMY, eygJ, Gggp), (2) our error analysis framework valuable and  insightful (ge5u, yUMY, eygJ), and (3) well-written and clearly presented (ge5u).  We summarize how we addressed the concerns below, and how the special situation affect our rebuttal:

## Note on reviewer reactions

Due to the bug on OpenReview this year, the scores are reverted back to the initial scores, but we want to highlight the reactions of the reviewers to our rebuttal:
- **Reviewer Gggp raised their score to 6** after acknowledging our updated results and detailed experiments.
- **Reviewer yUMY** stated that “the additional experiments, expanded ablations, and responses to the raised concerns help clarify several aspects of the work” and “find the rebuttal reasonable and the added experiments convincing”. However, the reviewer posted this response after the system was frozen and could not update the score accordingly.
- **Reviewer ge5u** followed up on our rebuttal with additional questions; we quickly replied, but the reviewer did not respond before the system was frozen. Overall, reviewers had positive views of our work, and their feedback indicates that we generally addressed their concerns besides a follow-up question on the additional results.

## Statistical significance (reviewer: ge5u, eygJ)

We added additional statistical analysis of our results and demonstrated statistically significant improvements in our framework SLIM over the baselines—the main experiments are run with three random seeds, and we show the mean and standard deviation in Table 13 of the updated PDF.

## Rigorous ablations (reviewer: ge5u, yUMY)

We added ablations over the key modules of our framework: the summarization module and the browse tool. These experiments justify our design decisions, and the reviewer yUMY found them “convincing”. The new ablation studies are in Table 10.

## Open-source and RL-trained models (reviewer: ge5u, yUMY)

We conducted experiments with open-weight and RL-trained models, GPT-OSS-120B and Tongyi-DeepResearch-30B, and found similar improvement with our framework SLIM over competitive baselines. Controlling for cost, we achieve significant improvements on BrowseComp. The added results are in Table 17 and 18.

## Discussion with existing literature (reviewer: ge5u, Gggp)

We thank the reviewers for providing additional works and references to compare to, and we have updated our Related Works (Section 7) and Additional Discussions (Appendix A.1) accordingly (changes are highlighted in red).

We sincerely appreciate the reviewers for engaging in discussions, and the area chair and program chairs for accommodating the special circumstances. We hope that our additional experiments and discussions are taken into consideration in the final decision.

---

### Meta-Review · Area_Chair_BmSw · 2026-01-07

**Summary:**

This paper addresses an important and timely challenge in long-horizon agentic search: managing context growth and tool inefficiency over extended trajectories. Reviewers generally found the paper to be clearly written, empirically solid within its chosen scope, and practically motivated. The proposed SLIM framework is simple and well-engineered, and the reported results show meaningful efficiency gains over several open-source baselines on BrowseComp and Humanity's Last Exam (HLE), with fewer tool calls and reduced hallucinations. The inclusion of a trajectory-level error taxonomy and automated analysis pipeline was viewed as a valuable diagnostic contribution that improves interpretability of agent failures. The authors’ rebuttal added ablations and multi-seed experiments that strengthened the empirical claims within the evaluated setting.

However, a key concern remains the disconnect between the generality of the paper’s claims and the narrowness of its empirical evaluation. The abstract and framing present SLIM as a broadly applicable solution for context management for long-horizon agentic search and deep research systems, yet the evaluation is restricted to only two benchmarks, both centered on web-based fact-finding with short, verifiable answers. These benchmarks do not adequately cover the diversity of long-horizon agentic search behaviors implied by the paper’s claims, such as varied trajectory structures, different information-seeking patterns, or alternative failure modes that arise across long-horizon settings. As a result, it is difficult to conclude that the proposed approach generalizes beyond the specific tasks evaluated.

Given this mismatch between the scope of the claims and the limited benchmark coverage, I recommend rejection. The work would be significantly strengthened by broader empirical validation across a wider range of long-horizon agentic search tasks that better align with the level of generality asserted in the paper’s motivation and abstract.

**Reviewer Concerns:**

Concerns addressed by the rebuttal
The rebuttal effectively addressed several methodological concerns raised during review. The authors added multi-seed experiments and variance reporting, resolving questions about statistical significance and result stability. They conducted substantially more rigorous ablation studies, clarifying the impact of summarization intervals, summarization strategies, and browse-tool design choices. Additional experiments on open-weight and RL-trained models, as well as the inclusion of HealthBench as a long-form generation task, helped strengthen the empirical evidence and partially addressed concerns about reliance on proprietary models and narrow task formats. Within the evaluated settings, the rebuttal materially improved clarity, rigor, and empirical support.

Concerns that remain outstanding
Despite these improvements, the core concern about evaluation breadth relative to the paper’s claims remains unresolved. While the addition of HealthBench is a positive step, the paper still evaluates SLIM on a very limited set of benchmarks given its framing as a general solution for long-horizon agentic search and long-context reasoning. If the primary claim is improved long-context summarization and long-horizon capability, the evaluation should extend to established benchmarks that directly stress these dimensions, such as multi-step reasoning and tool-use benchmarks (e.g., GAIA), long-context understanding benchmarks (e.g., LongBench), and multi-hop question answering benchmarks (e.g., HotpotQA). Without such evaluations, it remains unclear whether SLIM’s benefits generalize beyond a narrow subset of web-based information-seeking tasks. Thus, although the rebuttal strengthened rigor within the chosen scope, it did not sufficiently broaden the empirical validation to support the level of generality asserted in the abstract and motivation, which remains the primary basis for my recommendation to reject.

**Reviewer Scores:**

yUMY -> maintained the score since concerns were partially addressed
eygJ -> maintained the score since concerns were partially addressed
Gggp -> maintained the score since concerns were partially addressed
ge5u -> maintained the score since concerns were partially addressed

---

### Decision · Program_Chairs · 2026-01-26

Reject